# Oxygen isotopes suggest elevated thermometabolism within multiple Permo-Triassic therapsid clades

Kévin Rey[1,2]*, Romain Amiot[1], François Fourel[1,3], Fernando Abdala[2], Frédéric Fluteau[4], Nour-Eddine Jalil[5], Jun Liu[6], Bruce S Rubidge[2], Roger MH Smith[2,7], J Sébastien Steyer[5], Pia A Viglietti[2], Xu Wang[8], Christophe Lécuyer[1,9]

[1]Univ Lyon, Université Lyon 1, Ens de Lyon, CNRS, UMR 5276 LGL-TPE, Villeurbanne, France, France; [2]Evolutionary Studies Institute and School of Geosciences, University of the Witwatersrand, Johannesburg, South Africa; [3]Univ Lyon, Université Lyon 1, CNRS, UMR 5023 LEHNA, Villeurbanne, France, France; [4]Institut de Physique du Globe de Paris, Paris, France; [5]Centre de Recherches en Paléobiodiversité et Paléoenvironnements, UMR 7207 CNRS-MNHN-UPMC, Museum National d'Histoire Naturelle, Paris, France; [6]Key Laboratory of Vertebrate Evolution and Human Origins of Chinese Academy of Sciences, Institute of Vertebrate Paleontology and Paleoanthropology, Chinese Academy of Sciences, Beijing, China; [7]Iziko South African Museum, Cape Town, South Africa; [8]Key Laboratory of Cenozoic Geology and Environment, Institute of Geology and Geophysics, Chinese Academy of Sciences, Beijing, China; [9]Institut Universitaire de France, Paris, France

*For correspondence: kevin.rey@wits.ac.za

Competing interests: The authors declare that no competing interests exist.

**Abstract** The only true living endothermic vertebrates are birds and mammals, which produce and regulate their internal temperature quite independently from their surroundings. For mammal ancestors, anatomical clues suggest that endothermy originated during the Permian or Triassic. Here we investigate the origin of mammalian thermoregulation by analysing apatite stable oxygen isotope compositions ($\delta^{18}O_p$) of some of their Permo-Triassic therapsid relatives. Comparing of the $\delta^{18}O_p$ values of therapsid bone and tooth apatites to those of co-existing non-therapsid tetrapods, demonstrates different body temperatures and thermoregulatory strategies. It is proposed that cynodonts and dicynodonts independently acquired constant elevated thermometabolism, respectively within the Eucynodontia and Lystrosauridae + Kannemeyeriiformes clades. We conclude that mammalian endothermy originated in the Epicynodontia during the middle-late Permian. Major global climatic and environmental fluctuations were the most likely selective pressures on the success of such elevated thermometabolism.

## Introduction

One key adaptation enabling tetrapods to cope with fluctuating climatic conditions was the acquisition of endothermy (*Paaijmans et al., 2013*). This character is defined here as the ability to actively produce body heat through metabolic activity (*Cannon and Nedergaard, 2004*). Its development and anchoring in populations constitutes a major step in vertebrate evolution because it modified the energy relationships between organisms and their environments. By actively raising and maintaining body temperature within a narrow range that allows optimal physiological and biochemical

**eLife digest** School textbooks often refer to "cold-blooded" and "warm-blooded" animals, but these terms are misleading. Rather than being cold, animals like reptiles have body temperatures that are mostly determined by their external environment and can actually achieve high body temperatures, for example, by basking in the sun. By contrast, "warm-blooded" mammals produce their own heat and typically maintain a body temperature that is warmer than their environment. As such, so-called warm-blooded animals are more accurately referred to as "endotherms" and cold-blooded animals as "ectotherms".

Endothermic animals share several characteristics, including insulating layers – like fur or feathers – that keep the body warm, and a secondary palate that separates the mouth and nose for continuous breathing, even while eating. Many of these traits are seen in fossils belonging to a group of animals called the therapsids. Also known as the "mammal-like reptiles", these animals are descended from ectothermic reptiles but are the ancestors of the endothermic mammals. They dominated the land between 270 and 220 million years ago, during periods of time called the Permian and the Triassic. They also survived two major mass extinction events, including the most devastating mass extinction in all of Earth's history. However, when the ancestors of mammals became truly endothermic remains an open question. Previous studies that have tried to determine this by focusing on the physical characteristics of therapsids have not yet given a consistent date.

Rey et al. took a new approach to answer when endothermy first evolved in the mammal-like reptiles, and instead looked at the chemical makeup of minerals in over 100 fossils. Oxygen can exist in different forms called stable isotopes: oxygen-16 and the rarer and heavier oxygen-18. The ratio of these two isotopes in a fossil will depend on, among other things, where the animal lived and, importantly, its body temperature. Therefore, Rey et al. compared oxygen-containing minerals in the bones and teeth of therapsids to those of other animals that lived alongside them to look for signatures that indicated differences in body temperature and how it was regulated.

It appears that two different branches of the therapsid's family tree independently became endothermic. One branch includes the mammals and their direct ancestors, while the second is more distantly related to mammals. Both became endothermic towards the end of the Permian Period, between about 259 and 252 million years ago. Based on these findings, Rey et al. suggest that endothermy allowed these animals to better cope with fluctuating climates, which helped them to be among the few species that survived the mass extinction event at the end of the Permian.

Going forward, these new findings can help scientists to understand which physical characteristics were necessary for endothermy to first develop and which helped to optimize it afterwards. Furthermore, they also suggest that endothermic animals are more able to survive fluctuations in climate, which could guide efforts to protect modern-day endangered species that are most at risk from the ongoing effects of climate change.

functioning, endothermic vertebrates are able to colonise environments with extreme thermal conditions, for example freezing at high latitudes and altitudes (*Day et al., 2015*). Endothermy is commonly associated with homeothermy, being the capacity to regulate the body heat through metabolic activity as well. This combination corresponds to one end of a gradient of thermoregulatory strategies observed in living animals. The other end of the spectrum is ectothermy combined with poïkilothermy which animals use as a thermoregulatory strategy to increase their body temperature toward optimal levels by using external heat sources. Their body temperature therefore traces that of their surroundings and is the most commonly occurring energy saving strategy. Amongst modern vertebrates, various thermoregulatory strategies have been adopted between these two end-members, such as regional endothermy (*Bernal et al., 2001*; *Katz, 2002*) or inertial homeothermy (*McNab and Auffenberg, 1976*), and only birds and mammals fall within the endothermy end of the spectrum. It has been proposed that bird thermoregulation originated within non-avian dinosaurs (*Seebacher, 2003*; *Amiot et al., 2006*; *Grady et al., 2014*), or even earlier within basal archosauriforms (*Farmer and Carrier, 2000*; *de Ricqlès et al., 2003*; *Seymour et al., 2004*; *Summers, 2005*; *Gower et al., 2014*; *Legendre et al., 2016*). Various approaches have been tried by

many researchers to assess the origin of mammalian endothermy (*McNab, 1978*; *Bennett and Ruben, 1986*; *Farmer, 2000*; *Hillenius, 1992*, *1994*; *Kemp, 2006a*; *Khaliq et al., 2014*; *Owerkowicz et al., 2015*; *Benoit et al., 2016b*; *Crompton et al., 2017*). Some consider the appearance of endothermy to have either occurred during the transition from basal synapsid 'pelycosaurs' to therapsids, and to be either due to a shift in foraging ecology (*Hopson, 2012*) or due to a response to the availability of a seasonally arid, savanna-like biome by the end of the Early Permian (*Kemp, 2006b*).

How and why endothermy evolved in mammals remains a contentious issue, mostly because of the very low fossilization potential of anatomical and behavioural features associated with thermo-regulation. Amongst the latter features, the presence of hair as an insulating integument is unequivocally associated with endothermy in all extant mammals. The oldest synapsid fossils preserved with fur imprints are *Castorocauda* (*Ji et al., 2006*) and *Megaconus* (*Zhou et al., 2013*). These early relatives of mammals were recovered from the Middle-Late Jurassic of China, implying that hair and fur appeared before ~165 Ma. The occurrence of retracted, fully ossified and non-ramified infraorbital canals (a structure associated with the presence of maxillary vibrissae) within non-mammaliaform Prozostrodontia, implies an older age of approximately 240 to 246 Ma for the occurrence of fur and hair (*Benoit et al., 2016b*).

Another anatomical character interpreted as associated with endothermy is the bony secondary palate. This is a feature associated with efficient respiratory capabilities considered to be linked to the high energy required for elevated metabolic rates. In some Triassic non-mammaliamorph therapsids, bauriid therocephalians and cynodonts, a bony secondary palate is fully developed (*Abdala et al., 2014*). It is noteworthy that a complete bony secondary palate is also present in dicynodonts, however it is primarily formed by the premaxilla (*King, 1988*) and not the maxilla as documented in therocephalians, cynodonts and extant mammals. Although a secondary osseous palate is ubiquitous in mammals, it also occurs in a few ectotherms (crocodiles, scincid lizards), thus questioning its direct link to endothermy (*Bennett and Ruben, 1986*).

Almost all extant endotherms possess nasal turbinate bones covered with mucosa that reduce heat loss and moisten air during respiration (*Owerkowicz et al., 2015*). This feature, absent in extant ectotherms (*Witmer, 1995*), may have been present in therocephalian, cynodont and dicynodont therapsids, as postulated from bony ridges in the nasal cavities interpreted as supports for the turbinate complex (*Hillenius, 1992*, *1994*; *Crompton et al., 2017*).

A peculiar histological structure of fast-growing bone associated with highly vascularised woven-fibred matrix and primary osteons known as fibrolamellar bone (FLB), is another feature often used as evidence of a high metabolic activity (*Montes et al., 2010*; *Legendre et al., 2016*). Accordingly, several bone palaeohistological studies have addressed the quest for the presence of FLB in therapsids (*de Ricqlès, 1972*, *1979*; *Botha, 2003*; *Botha and Chinsamy, 2001*, *2004*; *Ray et al., 2004*; *Olivier et al., 2017*). *Ray et al. (2004)* and *Olivier et al. (2017)* analysed several therapsid groups (anomodont, gorgonopsian, therocephalian, cynodont) and found FLB in some genera (*Aelurognathus*, *Pristerognathus*, *Tritylodon*, *Oudenodon*, *Lystrosaurus*, *Moghreberia*), suggesting sustained fast growth, and thus elevated metabolic activity. The presence of FLB has also been demonstrated in some earlier non-therapsid synapsids such as *Sphenacodon*, *Dimetrodon* or even *Ophiacodon* (*Huttenlocker et al., 2010*; *Shelton et al., 2012*; *Shelton and Sander, 2017*). However, FLB also occurs in a few ectotherms such as in some turtles and crocodilians, and is absent in small mammals and passerine birds (*Bouvier, 1977*), showing that FLB is mostly correlated with high growth rates, which does not always correlate to high metabolic rates. Therefore, these characters alone cannot be considered as definitive evidence of endothermy, leaving the question of the timing and selection pressure for the origin of mammal endothermy still heavily debated.

Because the oxygen isotope fractionation between bone or tooth phosphate and body fluids is temperature dependent, and phosphate has a strong resistance to diagenetic alteration, oxygen isotope compositions of therapsid apatite phosphate ($\delta^{18}O_p$) has been used in this pilot study to investigate the origin of mammalian endothermy. Indeed the $\delta^{18}O_p$ value of vertebrate apatite (bone, tooth) reflects both the oxygen isotope composition of the animal body water ($\delta^{18}O_{bw}$) and its body temperature ($T_b$). Body water derives mainly from drinking meteoric water or plant water (*D'Angela and Longinelli, 1990*; *Kohn, 1996a*), and the $\delta^{18}O$ value of this water in turn depends on climatic parameters such as air temperature, hygrometry, and amount of precipitation (*Dansgaard, 1964*; *von Grafenstein et al., 1996*; *Fricke and O'Neil, 1999*).

Variations in the $\delta^{18}O$ values of ectotherm apatite, along with increasing latitude, are expected to reflect decreasing air temperatures as their body temperatures follow those of the environment. In contrast endotherms, which have a constant body temperature, should not be affected by environmental temperatures changes. Moreover, physiological adaptation to specific habitat use (aquatic, semi-aquatic or terrestrial) affects the $\delta^{18}O_{bw}$ value by controlling the magnitude of body input and output oxygen fluxes, some of them being associated with oxygen isotopic fractionations (*Amiot et al., 2010*). Therefore, co-existing endotherms and ectotherms should have distinct apatite $\delta^{18}O_p$ values reflecting their body temperature and ecological differences. By comparing apatite $\delta^{18}O_p$ values of therapsids with those of co-existing ectotherms of known ecologies at various palaeolatitudes, it should be possible to infer therapsid thermophysiology, a methodology that has previously been applied to non-avian dinosaurs (*Fricke and Rogers, 2000*; *Amiot et al., 2006*).

Following the protocol of previous research undertaken to establish the Permo-Triassic climatic conditions that prevailed during which South African tetrapods, including therapsids, radiated (*Rey et al., 2016*), this study aims to investigate thermophysiological strategies developed by various Permo-Triassic therapsid groups using the stable oxygen isotope compositions of their phosphatic remains. Our results add new data to the discussion of the origin of mammalian endothermy and its link to global climatic change.

## Results

### Permian therapsids

The 13 sampled South African Permian therapsids come from three different assemblage zones (AZ) of the Beaufort Group: the lower *Tapinocephalus* AZ, the *Tropidostoma* AZ and the lower *Daptocephalus* AZ (*Viglietti et al., 2016*).

Oxygen isotope compositions of three therapsid genera (*Dicynodon*, *Diictodon* and *Oudenodon*) from the youngest assemblage zone (lower *Daptocephalus* AZ) and seven therapsid genera (*Aelurosaurus*, *Diictodon*, *Ictidosuchoides*, *Oudenodon*, *Rhachiocephalus*, *Tropidostoma* and a basal cynodont) from the *Tropidostoma* AZ were respectively compared with one co-occuring *Rhinesuchus* and one co-occuring rhinesuchid. Differences between all the Permian therapsids and ectothermic temnospondyl range from $+1.1 \pm 0.6‰$ to $+8.0 \pm 0.9‰$ (*Figure 1A*), encompassing the expected range for which therapsids are considered ectothermic.

In addition, $\delta^{18}O_p$ values of the therapsids *Dicynodon*, *Diictodon* and *Oudenodon* are compared to those of the supposedly semi-aquatic parareptile *Pareiasaurus* (*Ivakhnenko, 2001*; *Kriloff et al., 2008*) (see Appendix 1), with an observed range of $+4.3 \pm 0.4‰$ to $+6.8 \pm 0.5‰$ (*Figure 1A*) which is similar to that measured between therapsids and amphibians. This also supports the ectothermic status of *Dicynodon*, *Diictodon* and *Oudenodon*.

*Anteosaurus*, *Criocephalosaurus*, *Struthiocephalus*, *Glanosuchus* and a titanosuchid from the lower *Tapinocephalus* AZ have also been compared to two co-occuring basal pareiasaurs which are attributed to either *Embrithosaurus*, *Nochelosaurus* or *Bradysaurus* (*Lee, 1997*) and are considered to have been terrestrial animals (*Canoville et al., 2014*). From *Figure 1B*, the $\delta^{18}O_p$ differences range from $-1.4 \pm 0.6‰$ to $0.7 \pm 1.0‰$, also supporting the ectothermic status of these therapsids.

From the Middle Permian of China, one anteosaurid *Sinophoneus yumenensis* from the low palaeolatitude locality of Dashankou has a $\delta^{18}O_p$ value $4.4 \pm 0.3‰$ lower than the co-existing bolosaurid parareptile *Belebey chengi*, which is considered to have been a terrestrial ectotherm (*Berman et al., 2000*; *Müller et al., 2008*). This difference between only two values would suggest that *Sinophoneus* was endothermic, but it is also very close to the expected ranges for ectothermic therapsids (*Figure 1B*). Considering *Sinophoneus* as semi-aquatic, as has been suggested for some anteosaurids (*Boonstra, 1955*, *Boonstra, 1962*), the $\delta^{18}O_p$ difference would imply an ectothermic thermophysiology for this therapsid. This hypothesis needs to be tested with a larger number of samples, which are not yet available.

### Early to Middle Triassic therapsids

From the *Cynognathus* AZ (subzone B) of South Africa, differences between the therapsids *Kannemeyeria*, *Cynognathus* and *Diademodon* and the temnospondyl amphibians *Xenotosuchus* and *Microposaurus* range from $-1.5 \pm 1.1‰$ to $+0.9 \pm 1.5‰$ (*Figure 2A*), which fit within the range

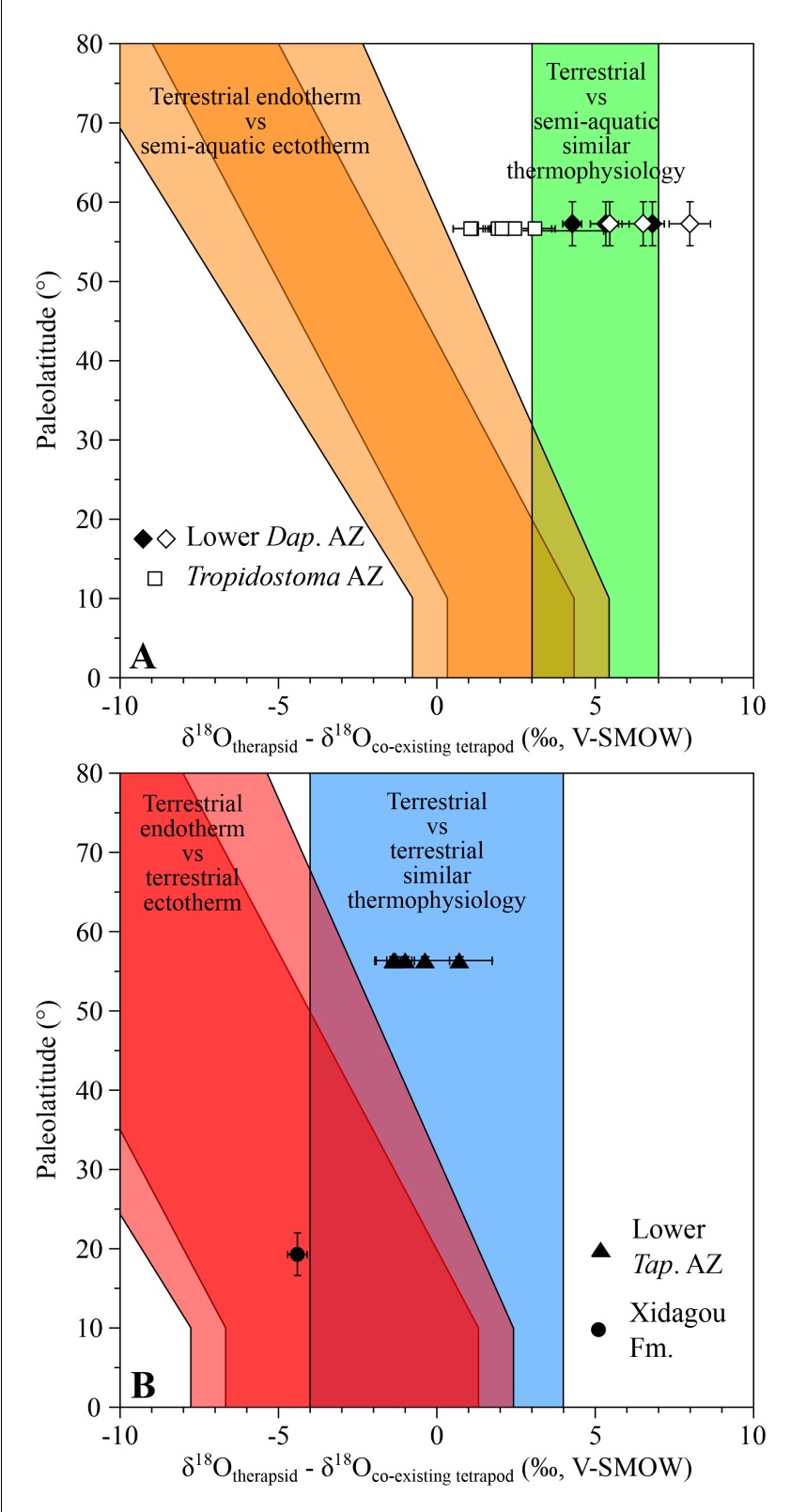

**Figure 1.** $\delta^{18}O_p$ differences between Permian therapsids and other tetrapods. Differences in $\delta^{18}O_p$ values between therapsids and stereospondyls (white symbols) and between therapsids and parareptiles (black symbols) from the same localities are plotted against their corresponding palaeolatitude. A theoretical framework based on modern temperature gradient (0.6 ± 0.1°C/°Lat; see Appendix 1) and phosphate-water-temperature

*Figure 1 continued on next page*

*Figure 1 continued*

oxygen isotope fractionation (*Lécuyer et al., 2013*) predicts various $\delta^{18}O_p$ value differences. The lighter orange and red areas correspond to the uncertainty of the temperature gradient. *Dap.: Daptocephalus; Tap.: Tapinocephalus.*

predicting endothermic therapsids. Interestingly, these therapsids have values ranging from 0.0 ± 1.6‰ to +1.8 ± 1.6‰ higher than the coexisting terrestrial archosauriform *Erythrosuchus* (*Botha-Brink and Smith, 2011*), a range suggesting that they shared a similar thermophysiology (*Figure 2B*). Therefore $\delta^{18}O_p$ values imply that, as in the case of the therapsids, *Erythrosuchus* was also endothermic which is consistent with the elevated growth rates implied by its palaeohistology (*de Ricqlès et al., 2008*; *Botha-Brink and Angielczyk, 2010*).

Also from South Africa, five *Lystrosaurus* specimens from the lower *Lystrosaurus* AZ have $\delta^{18}O_p$ values similar to those of the co-existing semi-aquatic stereospondyl *Lydekkerina* (*Schoch, 2008*; *Canoville and Chinsamy, 2015*). In addition, an indeterminate lystrosaurid from the Induan Jiu-caiyuan Formation of the Xinjiang Province has a $\delta^{18}O_p$ value similar (with a difference of −0.1 ± 0.6‰; *Figure 2B*) to that of the proterosuchid '*Chasmatosaurus*' *yuani*, a basal archosauriform considered terrestrial and possessing an intermediate thermometabolsim based on a palaeo-histological study (*Botha-Brink and Smith, 2011*). The combined results from South Africa and China suggest that the analysed lystrosaurids were terrestrial endotherms (*Figure 2*; see Appendix 1).

From the Ermaying Formation of the Shanxi Province (China), the therapsids *Shansiodon wangi* and *Parakannemeyeria youngi* have respectively $\delta^{18}O_p$ values of 2.0 ± 0.7‰ and 1.7 ± 0.7‰. These are both lower than the sampled erythrosuchid archosauriform *Shansisuchus shansisuchus*, which fall within two theoretical overlapping ranges (*Figure 2B*). As for the South African erythrosuchids, if we consider *Shansisuchus* as a terrestrial endotherm-like animal and the low palaeolatitude of this part of China region, then the two therapsids also fall within the range of endotherms.

## Middle to Late Triassic therapsids

The late Anisian cynodont *Diademodon* and the kannemeyeriiform *Kannemeyeria*, from the *Cynognathus* AZ (subzone C), have both lower $\delta^{18}O_p$ values than those of the contemporary semi-aquatic stereospondyls *Paracyclotosaurus* and *Xenotosuchus* with differences ranging from −3.9 ± 2.7‰ to −0.5 ± 0.6‰ (*Figure 3A*). This pattern fits within the main range predicting endothermic therapsids.

The Moroccan kannemeyeriiform *Moghreberia nmachouensis* from the early middle Carnian of the Argana Basin has a mean $\delta^{18}O_p$ value 2.0 ± 0.5‰ higher than the co-existing aquatic stereospondyl *Almasaurus habbazi* (*Figure 3A*), thus implying that *Moghreberia nmachouensis* was also endothermic.

An indeterminate cynodont from the Rhaetian Lower Elliot Formation of Lesotho has a $\delta^{18}O_p$ value 2.1 ± 0.3‰, higher than that of an indeterminate basal sauropodomorph. The suspected endo-thermy and terrestriality of both dinosaurs (*Amiot et al., 2006*; *D'Emic, 2015*) and cynodonts are in agreement with their $\delta^{18}O_p$ difference that falls within the expected range predicting similar thermo-physiology between the two (*Figure 3B*).

## Discussion

According to the $\delta^{18}O_p$ value differences observed between therapsids and co-existing non-therap-sid tetrapods, elevated thermometabolism seems to have been acquired by at least two therapsid clades: the unnamed dicynodont clade comprising Lystrosauridae + Kannemeyeriiformes, abbrevi-ated the 'L+K' clade, and the Eucynodontia (*Figure 4*).

Among the interpreted endothermic therapsids, six belong to the L+K clade (*Figure 4*): *Moghre-beria nmachouensis*, *Parakannemeyeria youngi*, *Kannemeyeria simocephalus* and *Shansiodon wangi* belong to the Kannemeyeriiformes clade, whereas the Lystrosauridae clade comprises *Lystrosaurus* and the Chinese indeterminate lystrosaurid. The interpretation of these taxa as endothermic-like ani-mals is better supported for the African Kannemeyeriiformes, *M. nmachouensis* and *K. simocephalus*, where more individuals were sampled, than for the Chinese species, *S. wangi* and *P. youngi*. Con-cerning the lystrosaurids, *Viglietti et al. (2013)* demonstrated aggregating behaviour in the Early

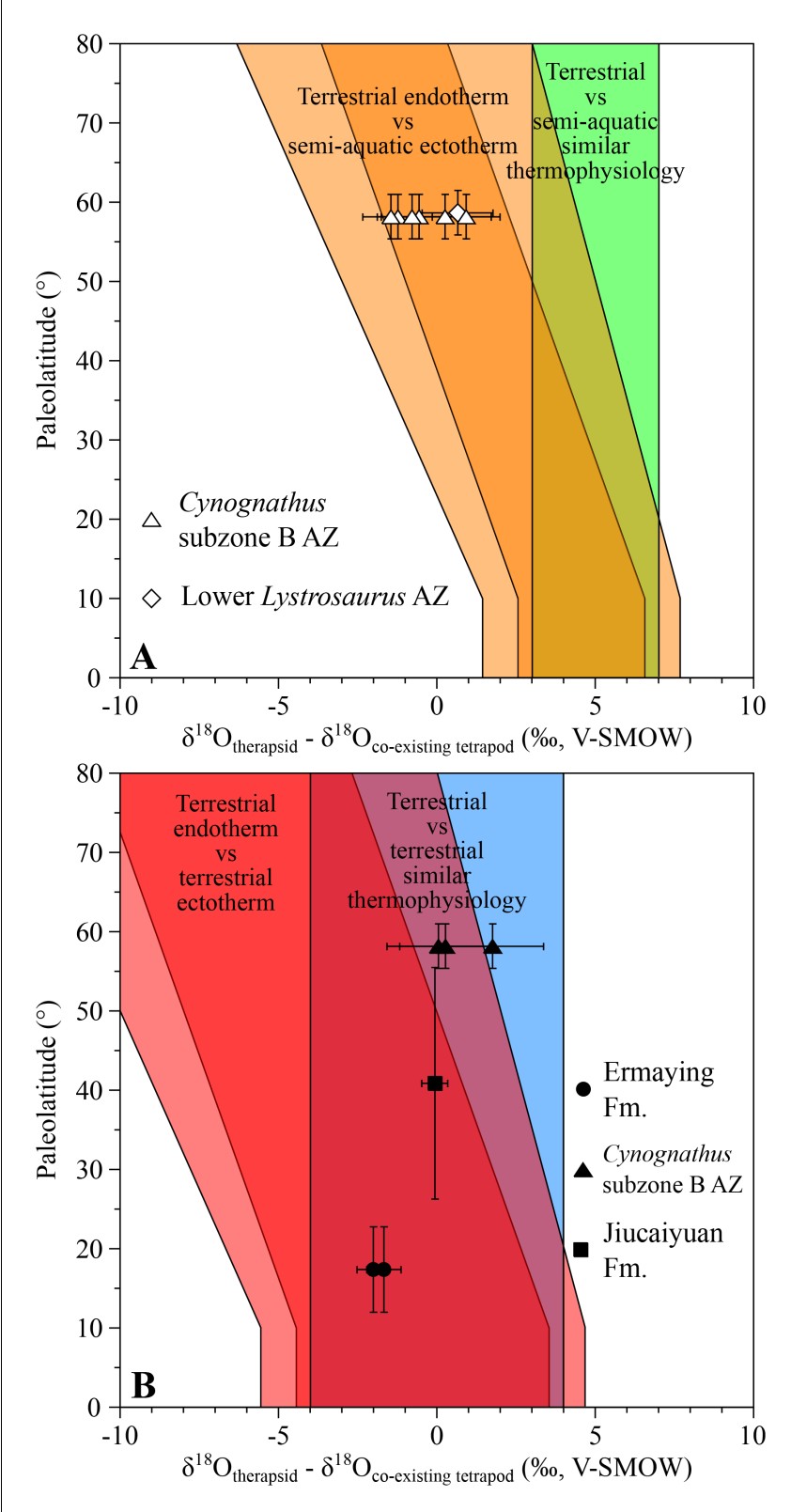

**Figure 2.** $\delta^{18}O_p$ differences between Early to Middle Triassic therapsids and other tetrapods. Differences in $\delta^{18}O_p$ values between therapsids and stereospondyls (white symbols) and between therapsids and archosauriforms (black symbols) from the same localities are plotted against their corresponding palaeolatitude. A theoretical framework based on a lower-than-today thermal gradient (0.4 ± 0.1°C/°Lat; see Appendix 1) and
*Figure 2 continued on next page*

*Figure 2 continued*

phosphate-water-temperature oxygen isotope fractionation (*Lécuyer et al., 2013*) predicts various $\delta^{18}O_p$ value differences. The lighter orange and red areas correspond to the uncertainty of the temperature gradient.

Triassic *L. declivis* and interpreted this as a means to keep warm under extreme climatic conditions. This is in agreement with our endothermic interpretation for the genus.

Based on the above interpretations, *Dicynodon* seems to have been ectothermic, a fact which would suggests the rise of endothermy amongst the dicynodont L+K clade during the latest Permian (Lopingian). This can be investigated in the future through the isotopic study of a basal dicynodontoid such as *Daptocephalus* from the Lopingian of South Africa (*Kammerer et al., 2011*; *Viglietti et al., 2016*) or *Peramodon* from the Salarevo Formation of Russia (*Sushkin, 1926*).

Based on our interpretations, the monophyletic group Eucynodontia (*Ruta et al., 2013*) (including *Cynognathus, Diademodon,* an indeterminate cynodont from Lesotho, and extant mammals) possessed endothermic-like metabolism. A parsimonious interpretation would imply rooting the origin of mammal endothermy within non-Eucynodontia Epicynodontia, between the end-Permian and the earliest Triassic. According to our results, the closest sampled relative of Eucynodontia, the late Permian basal cynodont (SAM-PK-K05339), was probably ectothermic. Therefore the origin of mammal endothermy could have taken place among 'intermediate' groups belonging to the Epicynodontia (such as *Cynosaurus* from the latest Permian or *Thrinaxodon* from the Early Triassic of South Africa) that have in the past been considered to have been endothermic, based on anatomical features (*Hillenius and Ruben, 2004*; *Benoit et al., 2015*, *2016a*).

In agreement with recent phylogenies (*Kemp, 2012*; *Ruta et al., 2013*), endothermic-like body temperature regulations seem to root sometime during the late Permian (Lopingian) independently within the L+K and Epicynodontia clades, the latter being at the origin of mammal endothermy.

An alternative hypothesis considers that both the L+K and Epicynodontia clades possessed a homologous endothermy inherited from their direct common ancestors, the basal Neotherapsida (*Figure 4*). This suggests that biochemical and physiological mechanisms enabling mammal endothermy, appeared at the base of the neotherapsids during the middle Permian, which is a conclusion recently published based on the paleohistology of some dicynodonts (*Olivier et al., 2017*). In our case, the absence of an endothermic signal in the sampled Permian therapsids could be due to an endothermy being only seasonal, linked to the presence of a cold season or to the reproduction period (as observed today in some reptile species; *Tattersall et al., 2016*), which would not be visible in a bulk signal. Therefore, effective acquisition of mammal 'true endothermy' was expressed independently within these two lineages, possibly as a result of extrinsic factors.

Global and regional palaeoclimate reconstructions show a cooling trend toward the end-Permian, followed by an abrupt and intense warming at the Permian-Triassic Boundary (*Chen et al., 2013*; *Rey et al., 2016*). Interestingly, most of the therapsid clades which survived the end-Permian mass extinction were supposedly endothermic. It thus appears that climatic fluctuations may have acted as selective pressures which favoured or 'activated' elevated thermometabolic capabilities within therapsids, at the origins of mammal endothermy. A possible explanation could be the acquisition of a fast growth rate due to the high metabolic rate of the endothermy. According to a recent palaeohistology study (*Botha-Brink et al., 2016*), Early Triassic therapsids, such as *Lystrosaurus* or even therocephalians and cynodonts, had a high growth rate allowing them to reach reproductive maturity within a few seasons and compensate their shortened life expectancies. This adaptation might have enabled certain therapsids to survive the intense climatic change of that time and conquer the newly vacant niches.

## Concluding remarks

In order to investigate the origin of mammal endothermy amongst the Permo-Triassic therapsids, stable oxygen isotope compositions of apatite phosphate and carbonate from therapsids and associated taxa recovered from several palaeolatitudes were analysed. The following results are highlighted:

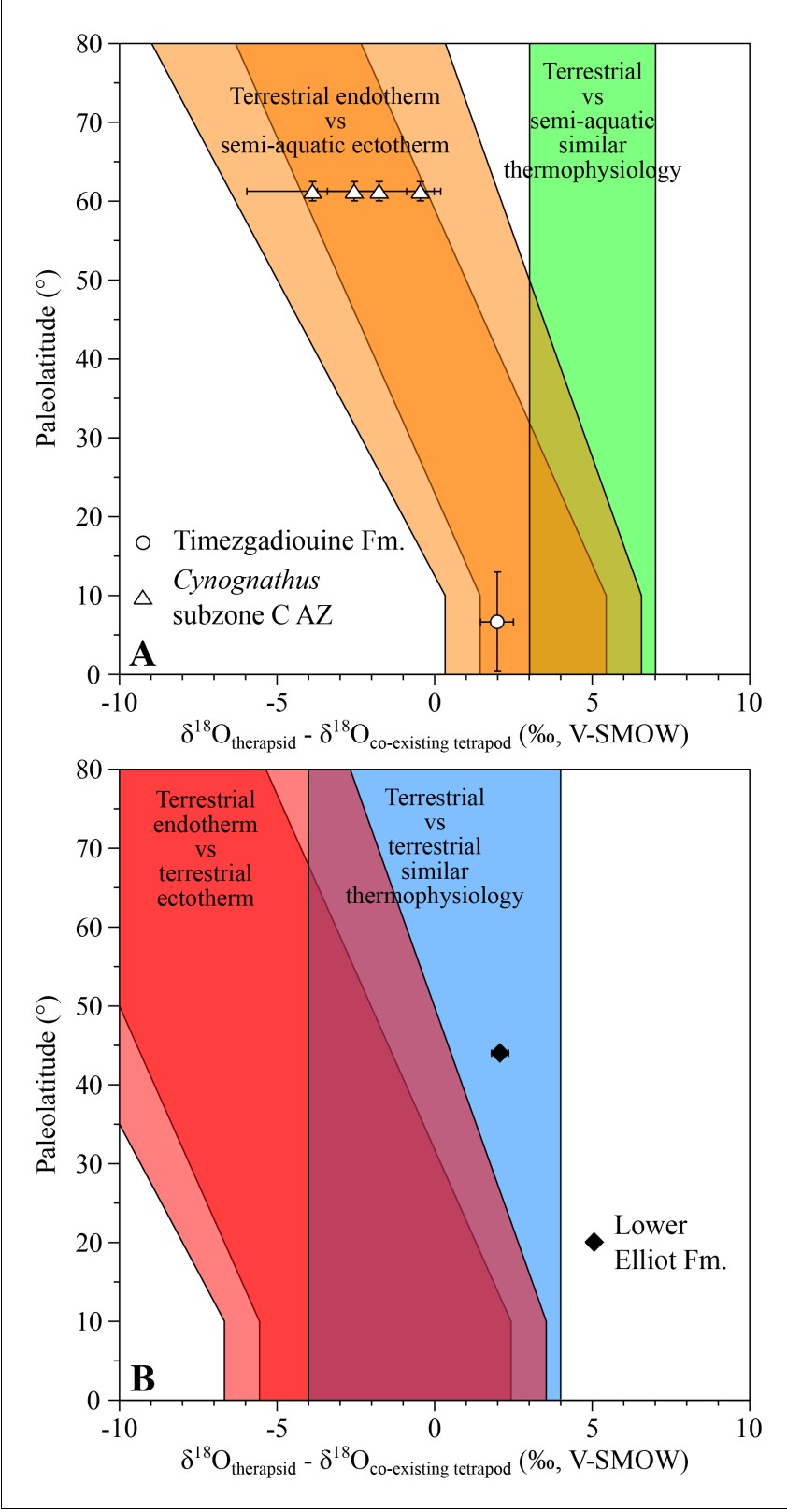

**Figure 3.** $\delta^{18}O_p$ differences between Middle to latest Triassic therapsids and other tetrapods. Differences in $\delta^{18}O_p$ values between therapsids and stereospondyls (white symbols) and between therapsids and archosauriforms (black symbols) from the same localities are plotted against their corresponding palaeolatitude. A theoretical framework based on a lower-than-today thermal gradient (0.5 ± 0.1°C/°Lat; see Appendix 1) and

*Figure 3 continued on next page*

*Figure 3 continued*

phosphate-water-temperature oxygen isotope fractionation (*Lécuyer et al., 2013*) predicts various $\delta^{18}O_p$ value differences. The lighter orange and red areas correspond to the uncertainty of the temperature gradient.

- Assuming that analysed samples have preserved their original isotope composition of phosphate, all the Permian therapsids analysed appear to have ectotherm-like thermoregulation and representatives of two Triassic therapsid clades are considered to have had endotherm-like thermoregulation: the Lystrosauridae + Kannemeyeriiformes and the Eucynodontia.
- It is proposed that constant elevated thermometabolism appeared independently, at least twice during therapsid evolution. Following the principles of parsimony and phylogenetic systematics, both evolutionary events occured during the late Permian.
- It seems that the timing of the acquisition of elevated thermometabolism among amniotes coincides with major global climatic and environmental fluctuations at the Permo-Triassic boundary and may had a selective advantage to survive the extinction event and result ultimately in mammalian endothermy.

## Material and methods

### Sample collection

Nineteen new fossil apatite samples were analysed to determine stable oxygen isotope compositions of apatite phosphate and carbonate, along with 89 samples for which oxygen isotope compositions have already been published (*Rey et al., 2016*; *Supplementary file 1*). This sample total comprises 41 teeth and 65 bones of 90 individual tetrapods (Therapsida, Archosauriformes, Parareptilia and Stereospondyli) recovered from Permian and Triassic deposits of South Africa, Lesotho, Morocco and China. All the sample localities are correlated to the marine biostratigraphic stages using the absolute ages accepted by the International Commission on Stratigraphy (*Cohen et al., 2013*; updated 12/2016), with the Permo-Triassic and Guadalupian-Lopingian boundaries now respectively considered to be at 251.90 ± 0.02 Ma (*Burgess et al., 2014*) and 259.1 ± 0.5 (*Zhong et al., 2014*) Ma.

South African samples comprise Permian and Triassic bones and teeth of therapsids, pareiasaurs, archosauriforms and stereospondyls recovered from 10 localities in the Beaufort Group (Karoo Supergroup), and housed in the collections of the Iziko South African Museum, Cape Town (SAM, *Supplementary file 1*) and at the Evolutionary Studies Institute, University of the Witwatersrand, Johannesburg (ESI, *Supplementary file 1*). Permian biozone ages of South African localities were taken from (*Rubidge et al., 2013*; *Day et al., 2015*), whereas Triassic age determination has been achieved by biostratigraphic correlation with Laurasian sequences (*Hancox et al., 1995*; *Rubidge, 2005*; *Abdala and Ribeiro, 2010*).

Lesotho samples comprise a cynodont therapsid and a basal sauropodomorph dinosaur from a Triassic locality near the town of Pokane, and are part of the Paul Ellenberger Collection at the Institut des Sciences de l'Evolution, University Montpellier, France (ISEM, *Supplementary file 1*). The locality belongs to the 'Red Beds inférieurs a or b' of the lower Elliot Formation which is currently regarded as latest Triassic (late Rhaetian) (*Knoll, 2004*).

Moroccan samples comprise therapsid and stereospondyl bones recovered from the 'Locality 11' of the Argana Group (*Jalil, 1999*) near the village of Alma, and housed at the Museum National d'Histoire Naturelle, Paris, France (MNHN, *Supplementary file 1*). The locality is biostratigraphically correlated to the upper Timezgadiouine Formation, considered to be Middle to early Late Carnian (*Jalil, 1999*).

Chinese samples are from Permian and Triassic localities situated in Gansu, Shanxi and Xinjiang provinces and comprise therapsids found in association with archosauriforms or parareptiles. These remains are curated at the Institute of Vertebrate Paleontology and Paleoanthropology in Beijing, China (IVPP, *Supplementary file 1*). The Dashankou locality, from Gansu Province, is biostratigraphically dated as Early Roadian (*Liu et al., 2009*; *Liu, 2010*). From Shanxi Province, sampled fossils originate from three localities in the Ermaying Formation which is considered to be Anisian (*Liu et al.,*

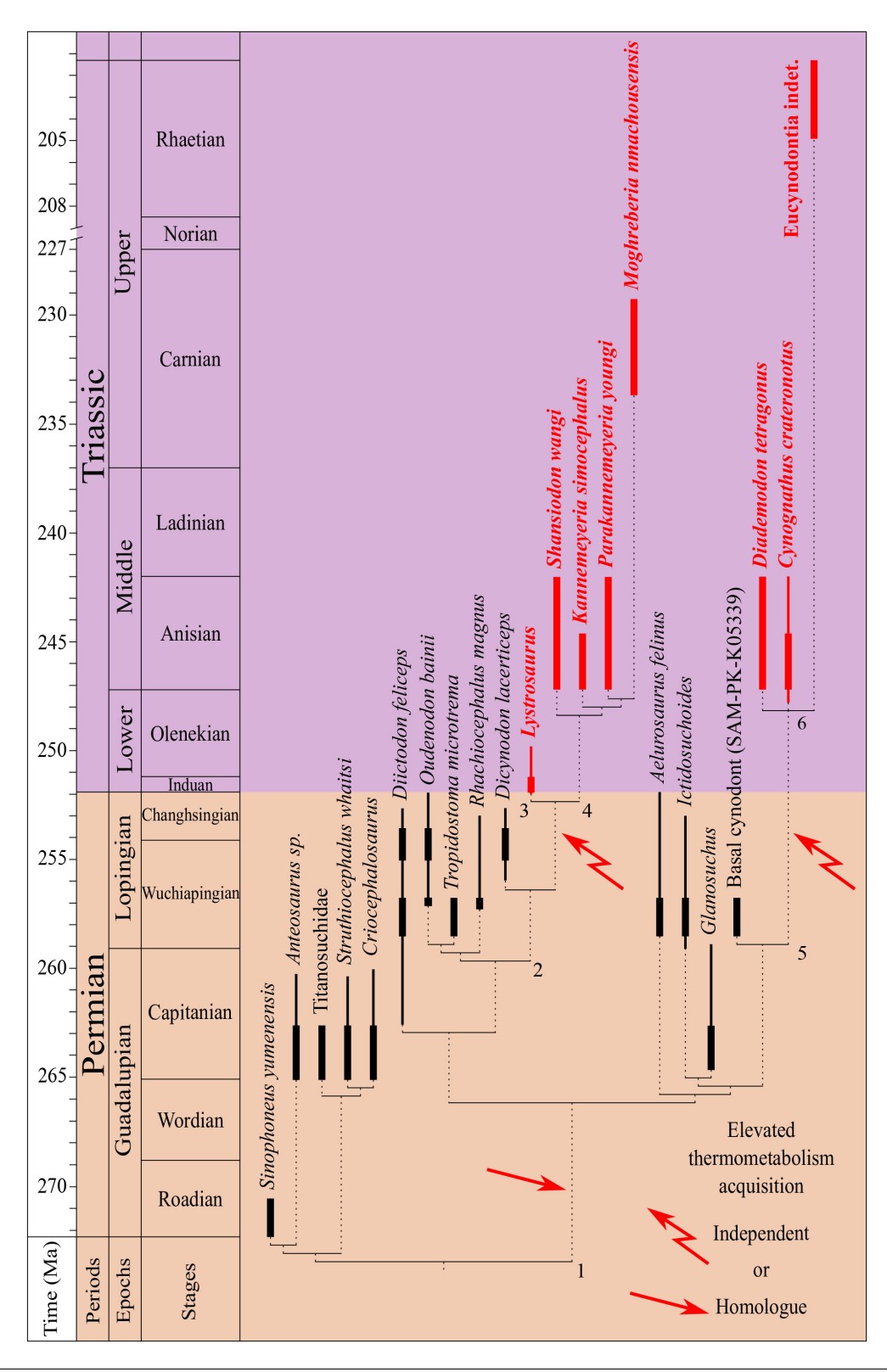

**Figure 4.** Phylogeny of sampled therapsids. Phylogeny of the sampled therapsids plotted alongside a stratigraphic scale, based on proposed therapsid phylogenies (**Ruben and Jones, 2000**; **Hillenius and Ruben, 2004**; **Gebauer, 2007**; **Cisneros et al., 2012**; **Liu, 2013**; **Ruta et al., 2013**) and their biostratigraphic ranges (**Kammerer et al., 2011**; **Kemp, 2012**; **Ruta et al., 2013**; **Huttenlocker, 2014**; **Day et al., 2015**; **Viglietti et al., 2016**). The thickest parts of the bold lines represent the age range uncertainty of the localities where the samples come from. Species identified as endotherm-like

*Figure 4 continued on next page*

Figure 4 continued
are written in bold and red. Node numbers refer to clades quoted in the text: 1: Neotherapsida; 2: Dicynodontoidea; 3: Lystrosauridae; 4: Kannemeyeriiformes; 5: Epicynodontia; 6: Eucynodontia.

*2013*). From Xinjiang Province, two localities in the Jiucaiyuan Formation have been sampled and are considered Early Triassic (*Metcalfe et al., 2009*).

Calculation of palaeogeographic coordinates of the sampling sites was performed after careful selection of the magnetic poles of West Gondwana (*Muttoni et al., 2001*), North China ( *et al., 1992*), South Jungar (*Choulet et al., 2013*) and the Alashan terrane (*Meng, 1992*; *Yuan and Yang, 2015*). The Apparent Polar Wander Path (APWP) of South Africa (*Torsvik et al., 2012*) was used to

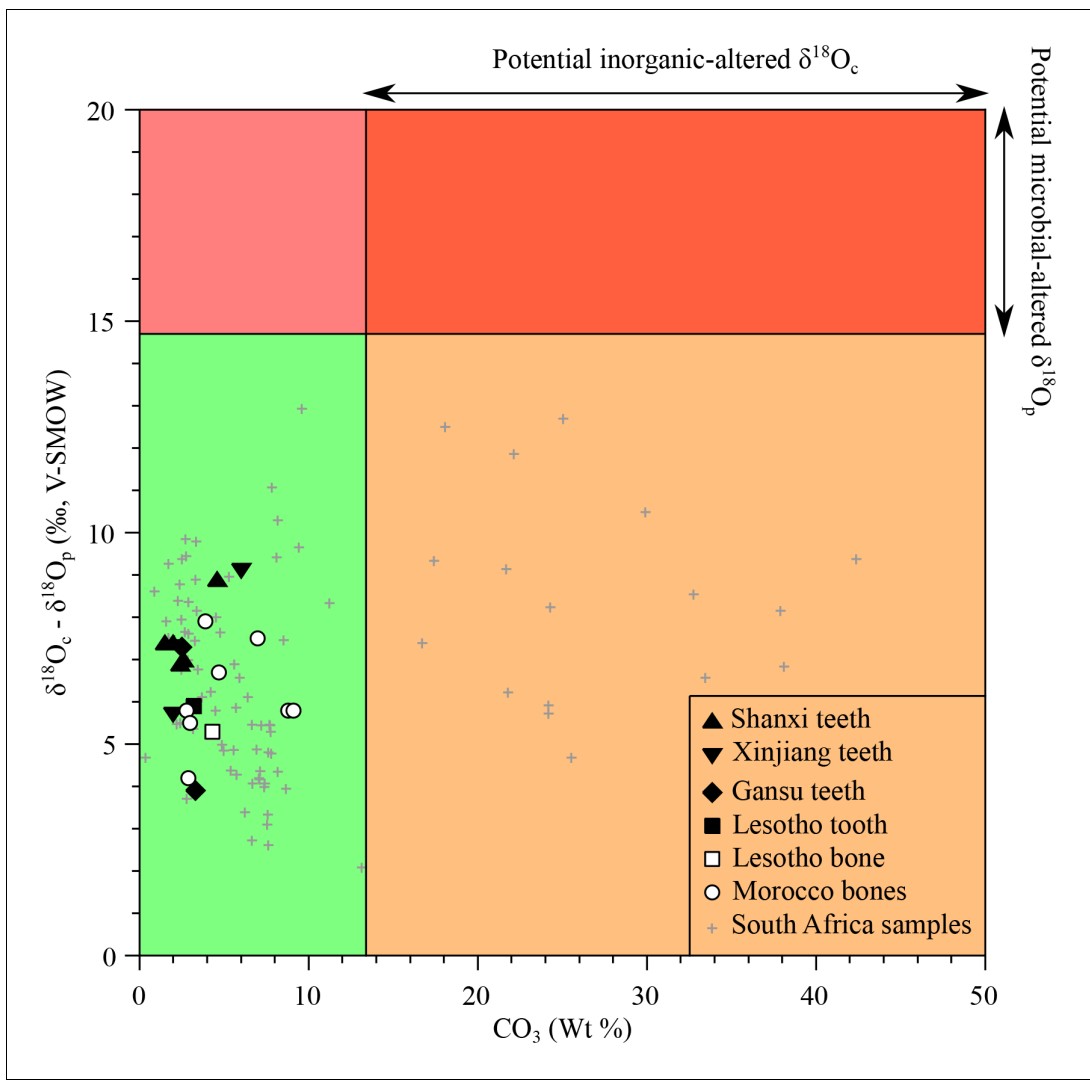

**Figure 5.** Isotopic preservation assessment. $\delta^{18}O_c$-$\delta^{18}O_p$ differences between teeth and bones plotted against the structural carbonate content (wt%) of apatite. Samples that have $\delta^{18}O_c$-$\delta^{18}O_p$ differences higher than 14.7‰ are considered doubtful as regards potential diagenetic alteration (see text). For carbonate contents (wt%) higher than 13.4%, the $\delta^{18}O_c$ values are considered to be inherited from inorganic diagenetic processes. A high difference between $\delta^{18}O_c$ and $\delta^{18}O_p$ is interpreted as the result of a microbially-mediated alteration of the apatite phosphate or too high $\delta^{18}O_c$ values resulting from the addition of inorganic carbonate or isotopic exchange with an external source of inorganic carbon. The grey crosses refer to previously published South African bone and tooth samples (*Rey et al., 2016*).

constrain the palaeolatitudinal position of the South African and Lesotho fossil sites. Palaeolatitudes and associated uncertainties (A95) are shown in *Supplementary file 1*.

## Analytical techniques

To measure the oxygen isotope composition of the apatite phosphate group, the phosphate ions were isolated using acid dissolution and anion-exchange resin applying a standard protocol (*Lécuyer, 2004*). Silver phosphate was quantitatively precipitated in a thermostatic bath set at a temperature of 70°C. After filtration, washing with double deionized water and drying at 50°C, an aliquot of 300 µg of $Ag_3PO_4$ was mixed with 300 µg of nickelised carbon in a silver reaction capsule. Silver phosphate was then reduced into CO to measure its $^{18}O/^{16}O$ ratio (*Lécuyer et al., 2007*; *Fourel et al., 2011*). Each sample was heated at 1450°C by pyrolysis using a VarioPYROcube EA system (Elementar) interfaced to an IsoPrime isotope ratio mass spectrometer working in continuous flow mode at the UMR CNRS 5276 LGLTPE, University Claude Bernard Lyon 1.

Isotopic compositions are quoted in the standard δ notation relative to V-SMOW. Silver phosphate precipitated from standard NBS120c (natural Miocene phosphorite from Florida) was repeatedly analysed ($\delta^{18}O = 21.71 \pm 0.20‰$; n = 30) along with the silver phosphate samples derived from the tetrapod remains. For the oxygen isotope analysis of apatite carbonate, about 10 mg of tooth or bone powder was pre-treated (*Koch et al., 1997*). Powders were washed with a 2% NaOCl solution to remove organic matter, then rinsed five times with double deionized water and air-dried at 40°C for 24 hr. Potential secondary carbonate was removed by adding 0.1 M acetic acid and leaving for 24 hr, after which the powder was again rinsed five times with double deionized water and air-dried at 40°C for 24 hr. The powder/solution ratio was kept constant at 0.04 g mL$^{-1}$ for both treatments. Stable isotope ratios were determined by using a Thermo Finnigan Gasbench II at the geochemistry laboratory of the Institute of Geology and Geophysics (Chinese Academy of Sciences, China). For each sample, an aliquot of 2 mg of pre-treated apatite was reacted with 5 drops of supersaturated orthophosphoric acid at 72°C for one hour under a He atmosphere before starting 10 measurement cycles of the isotopic composition of the $CO_2$ produced with a Finnigan MAT 253 continuous flow isotope ratio mass spectrometer. The measured oxygen isotopic compositions were normalized relative to the NBS-19 calcite standard and have a reproducibility index better than ±0.2‰. Isotopic compositions are quoted in the standard δ notation relative to V-SMOW.

## Robustness of the stable isotope record

Analysed materials consist of bone or tooth dentine, which is more porous than enamel with small and less densely inter-grown apatite crystals (*Mills, 1967*). Thus, their original stable isotope compositions are more prone to diagenetic alteration that may have taken place through precipitation of secondary minerals within and at the surface of bioapatite crystals, adsorption of ions on the surface of apatite crystals, or dissolution and recrystallization with isotopic exchange. The samples from South Africa have been previously tested for primary preservation through comparison between their $\delta^{18}O_p$ values, $\delta^{18}O_c$ values and carbonate content on the basis of the following considerations: (1) the carbonate content in apatite of modern vertebrates typically ranges from less than 1% up to 13.4%. Thus, samples that have a carbonate content exceeding 13.4 wt% likely contain additional inorganic carbonate precipitated from diagenetic fluids, and would result in potentially biased $\delta^{18}O_c$ values of apatite carbonate (*Figure 5*); (2) In modern vertebrates, the oxygen isotope composition of apatite carbonate is higher than that of co-occurring apatite phosphate (7–9 ‰ in mammals), and up to 14.7‰ in sharks (*Vennemann et al., 2001*). Experimental ( *et al., 1967*) and empirical studies (*Zazzo et al., 2004b*) have shown that microbially-mediated diagenetic alteration of apatite phosphate results in a greater difference between $\delta^{18}O_c$ and $\delta^{18}O_p$ values. Therefore, fossil samples exhibiting $\delta^{18}O_c$-$\delta^{18}O_p$ differences larger than 14.7‰ are most likely altered and can be disregarded (*Figure 5*). Inorganic alteration at low temperature has little effect on the $\delta^{18}O_p$ values of phosphates, even at geological time scales (*Lecuyer et al., 1999*), so samples affected by inorganic diagenetic alteration of carbonates, (resulting either in a high overall carbonate content or anomalous $\delta^{18}O_c$-$\delta^{18}O_p$ differences), may still preserve the original oxygen isotope composition of their phosphate (*Figure 5*). Using these two assessments, newly measured $\delta^{18}O_p$ values are considered to have preserved their original isotopic signatures and can be interpreted in terms of ecologies and physiologies.

## Assessment of therapsid thermophysiology

For all localities, average $\delta^{18}O_p$ values were calculated for each tetrapod species. Differences in $\delta^{18}O_p$ values between therapsid species and co-occurring non-therapsid tetrapods (amphibians, parareptiles or archosauriforms) were calculated and plotted against their corresponding palaeolatitude for three time intervals: the middle to late Permian (*Figure 2*), the Early to Middle Triassic (*Figure 3*) and the Middle Triassic to latest Triassic (*Figure 4*). These differences were compared to the following four theoretical areas of values represented as coloured areas in *Figures 1–3*. To construct those theoretical areas, both the phosphate-water temperature scale from *Lécuyer et al., 2013* and the differences of stable oxygen compositions between mammals of various ecologies from *Cerling et al. (2008)* have been used. (see Appendix 1 for their construction details).

Orange and green areas in *Figures 1A*, *2A* and *3A* represent expected $\delta^{18}O_p$ value differences between terrestrial therapsids and semi-aquatic stereospondyls (white symbols) or parareptiles (black symbols); red and blue areas in *Figures 1B*, *2B* and *3B* represent expected $\delta^{18}O_p$ value differences between terrestrial therapsids and terrestrial Permian parareptiles or Triassic archosauriforms (black symbols). Oblique orange and red areas in *Figures 1–3* represent expected $\delta^{18}O_p$ value differences between an endotherm and an ectotherm. Vertical green and blue areas in *Figures 1–3* represent expected $\delta^{18}O_p$ value differences between animals having similar thermophysiology.

## Acknowledgements

The authors thank the MNHN, the IVPP, Iziko SA Museum and the ESI for granting access to the fossils, and the South African Heritage Resources Agency (SAHRA) for their authorizations to sample and export fossils for stable isotope analysis (PermitID: 1858). We also acknowledge S Jiquel (ISEM) for the access to the Ellenberger's collection of the University Montpellier 2, J Falconnet (MNHN) for his identification, L. Cui (IGGCAS) for laboratory help and J Cubo (UPMC) for constructed discussions that improved the manuscript. This work was supported by a French project INSU of the CNRS, the Palaeontological Scientific Trust (PAST) and its Scatterlings of Africa programmes, National Research Foundation (NRF) and DST/NRF Centre of Excellence in Palaeosciences. AR and WX were supported by the National Basic Research Program of China (2012CB821900), and LJ by the National Basic Research Program of China (2012CB821902). We thank the four reviewers, K Angielczyk, J Eiler and two anonymous ones, for their constructive comments.

## Additional information

### Funding

| Funder | Grant reference number | Author |
|---|---|---|
| Institut National des Sciences de l'Univers, Centre National de la Recherche Scientifique | | Romain Amiot |
| Palaeontological Scientific Trust | | Bruce S Rubidge |
| National Research Foundation | | Bruce S Rubidge |
| Centre of Excellence in Palaeosciences | | Bruce S Rubidge |
| National Basic Research Program of China | 2012CB821900 | Romain Amiot Xu Wang |
| National Basic Research Program of China | 2012CB821902 | Jun Liu |
| Institut Universitaire de France | | Christophe Lécuyer |

The funders had no role in study design, data collection and interpretation, or the decision to submit the work for publication.

## Author contributions
KR, Conceptualization, Data curation, Formal analysis, Investigation, Writing—original draft, Project administration, Writing—review and editing; RA, Conceptualization, Data curation, Formal analysis, Supervision, Project administration, Writing—review and editing; FFo, Conceptualization, Data curation, Project administration, Writing—review and editing; FA, N-EJ, JL, RMHS, JSS, PAV, Resources, Writing—review and editing; FFl, Formal analysis, Investigation, Writing—review and editing; BSR, Resources, Funding acquisition, Project administration, Writing—review and editing; XW, Resources, Data curation, Formal analysis, Writing—review and editing; CL, Conceptualization, Data curation, Formal analysis, Supervision, Funding acquisition, Project administration, Writing—review and editing

## Author ORCIDs
Kévin Rey, http://orcid.org/0000-0002-6788-2453

## Additional files

### Supplementary files
• Supplementary file 1. Stable oxygen isotope compositions of phosphate ($\delta^{18}O_p$) and carbonate ($\delta^{18}O_c$) of Permo-Triassic tetrapod teeth and bones reported along with their stratigraphic position, estimated age, palaeolatitudes and their carbonate content. Asterisks represent diagenetically altered values.

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

## Appendix 1

### Theoretical curves construction

In order to infer therapsid thermophysiologies based on the $\delta^{18}O_p$ differences between therapsids and the associated fauna, the following theoretical framework is proposed and represented in *Figures 1*, *2* and *3* as colored areas. These areas are based on the two main factors influencing animal $\delta^{18}O_p$ values: their thermometabolism (here simplified as ectotherm and endotherm) and their lifestyle (here terrestrial and semi-aquatic).

The $\delta^{18}O_p$ value ranges are estimated based on the phosphate-water temperature scale (*Lécuyer et al., 2013*):

$$T_b = 117.4 - 4.5(\delta^{18}O_b - \delta^{18}O_{bw}) \tag{1}$$

Where $T_b$ corresponds to body temperature, $\delta^{18}O_p$ correspond to the oxygen isotope composition of apatite phosphate, and $\delta^{18}O_{bw}$ correspond to the oxygen isotope composition of body fluids. For endothermic vertebrates, $T_b$ is assumed to be at 37°C, the average $T_b$ of most extant placentals and possibly of the common ancestor of all extant mammals (*Watson and Graves, 2013*). For ectothermic vertebrates, $T_b$ is assumed here to reflect immediate environmental temperature ($T_e$). According to *Equation 1*, the $\delta^{18}O_p$ difference between an endotherm and an ectotherm can be expressed as:

$$\delta^{18}O_{p-endotherm} - \delta^{18}O_{p-ectotherm} = (T_e - 37)/4.5 + (\delta^{18}O_{bw-endotherm} - \delta^{18}O_{bw-ectotherm})$$

Because vertebrate $\delta^{18}O_{bw}$ value depend on their ecology that affects the input-output balance of body water (*Luz et al., 1984*; *Bryant and Froelich, 1995*; *Kohn, 1996a*), the difference $\delta^{18}O_{bw-endotherm} - \delta^{18}O_{bw-ectotherm}$ will mainly reflects that of their lifestyle (terrestrial, semi aquatic or aquatic), as well as their dependency on the surface water they ingest. *Cerling et al., 2008*, the $\delta^{18}O_p$ difference between water-dependent and water-independent terrestrial mammals can be up to 4‰. This is illustrated by the ranges 1 and 3 in *Appendix 1— Figure 1* and by the red and blue ranges in the main text *Figures 1A, 2A* and *3A*).

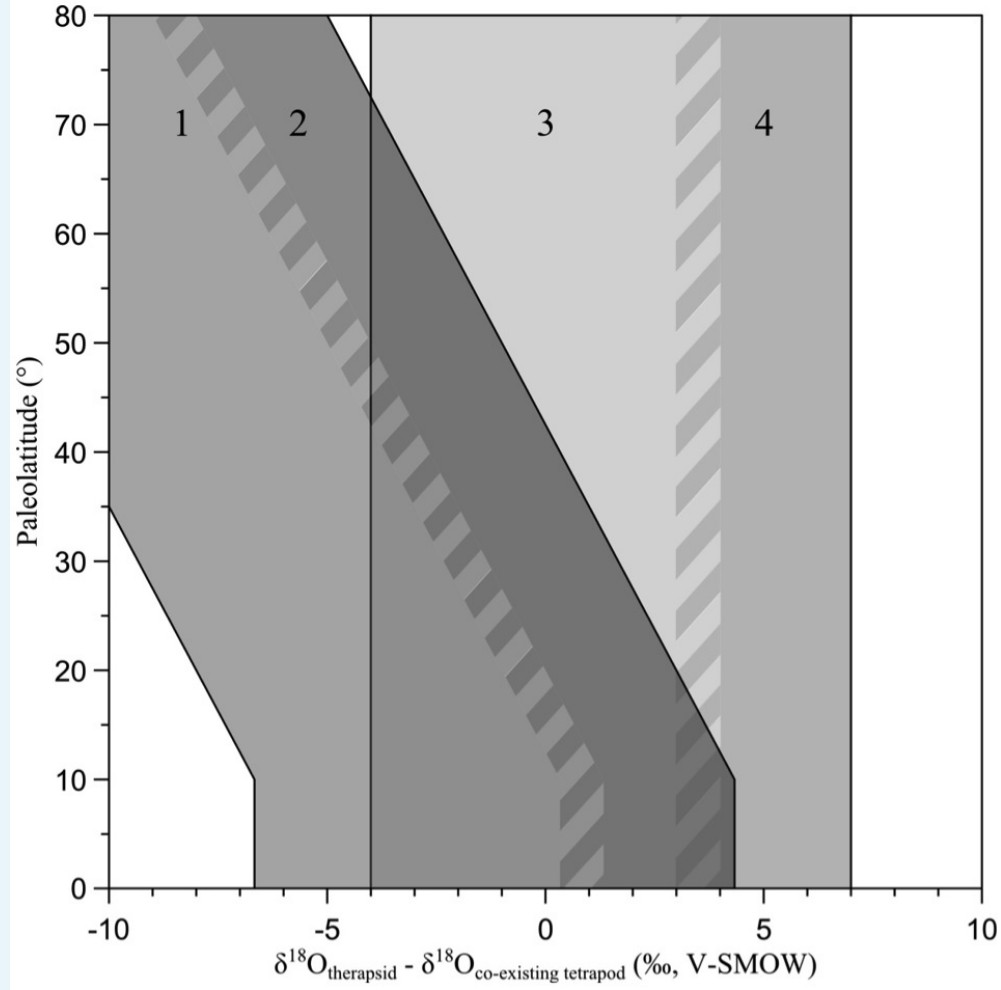

**Appendix 1—figure 1.** Expected latitudinal variation of the $\delta^{18}O_p$ difference between vertebrate taxa of various physiologies and ecologies. Based on modern relationships between climate and phosphate-water-temperature oxygen isotope fractionation (*Lécuyer et al., 2013*) the following $\delta^{18}O_p$ values differences are predicted: (1) corresponds to a terrestrial endotherm compared to a terrestrial ectotherm; (2) corresponds to a terrestrial endotherm compared to a semi-aquatic ectotherm; The vertical range (3) corresponds to terrestrial animals having similar thermometabolism; (4) corresponds to the difference between a terrestrial and a semi-aquatic animals having a similar thermometabolism.

This variation in $\delta^{18}O_p$ values can be overprinted by the animal lifestyle, a semi aquatic animal having $\delta^{18}O_p$ value 3‰ lower than that of co-existing water-dependent terrestrial species, and 7‰ lower than that of co-existing water-independent terrestrial one (*Appendix 1—figure 1*, ranges 2 and 4, and main text *Figures 1B*, *2B* and *3B*, ranges orange and green).

The environmental temperature ($T_e$) used to approximate ectotherm body temperature ($T_b$) is estimated based on the present-day relationship between mean air temperature and latitude. Because this latitude-temperature relationship is not valid for low latitudes below about 10° (corresponding to the thermal equators of present-day Earth), we assume a constant temperature between 0° and 10° of latitude.

As a simplification, three periods are considered:

1. The Middle to Late Permian with equatorial sea surface temperatures close to the modern ones (*Chen et al., 2013*) (~25°C) for which we assume a present-day thermal gradient of 0.6 °C/°Latitude (*Amiot et al., 2004*; *Williams et al., 2007*) (Main text *Figure 1*).
2. The Early to Middle Triassic having globally warmer temperatures and a flatter temperature gradient. Based on (*Trotter et al., 2015*), mean equatorial sea surface temperatures were about 10°C higher than Late Permian ones (*Trotter et al., 2015*) (~35°C). For this time period, we assume a flatter thermal gradient that we arbitrarily set at 0.4 °C/°Latitude (Main text *Figure 2*).
3. The Middle Triassic to Early Jurassic having globally intermediates temperatures (*Trotter et al., 2015*) that we set to 30°C with an intermediate gradient of 0.5 °C/°Latitude (Main text *Figure 3*).

In order to take into account the possible variations of the thermal gradient within the selected periods, an interval of ±0.1 °C/°L is added for each Main text figures.

Consequently, latitudinal thermal gradients will lead to $\delta^{18}O_{p\text{-endotherm}}$ - $\delta^{18}O_{p\text{-ectotherm}}$ differences varying along with latitude (*Appendix 1—figure 1*, ranges 1 and 2). This simplified framework is used in this study to predict the following scenarios based on $\delta^{18}O_p$ differences between therapsids and co-existing other tetrapods (*Appendix 1—figure 1*): Endothermic and terrestrial therapsid vs ectothermic and terrestrial tetrapod (range 1); Endothermic and terrestrial therapsid vs ectothermic and semi-aquatic tetrapod (range 2); Therapsids and other tetrapods having similar thermophysiologies and lifestyle (range 3); Terrestrial therapsid and semi-aquatic tetrapods having similar thermophysiologies (range 4).

## Ecology of sampled tetrapods and its impact on stable oxygen isotope compositions

Prior to interpreting differences between therapsid and associated non-therapsid tetrapod $\delta^{18}O_p$ values in terms of differences in thermophysiologies, ecological traits must be considered as they also affect the oxygen isotope compositions of apatite phosphate. Indeed, $\delta^{18}O_p$ values recorded in phosphatic tissues depend on the animal body temperature, as well as on the oxygen isotope composition of body water, the latter being affected by water turnover rate and isotopic fractionations associated with water loss (*Bryant and Froelich, 1995*). Indeed, water loss through transcutaneous evaporation, sweat and exhaled water vapour tends to $^{18}O$-enrich the remaining body water (*Kohn, 1996a*). This isotopic enrichment is amplified or mitigated depending on the rate of water turnover, which depends itself on the animal ecology. Aquatic and semi-aquatic animals regularly ingest and release large amounts of their environmental water compared to terrestrial ones, which in turn reduces the magnitude of body water $^{18}O$-enrichment relative to that of environmental water (*Kohn, 1996a*).

### Stereospondyl amphibians

The sampled stereospondyl amphibian clade (*Almasaurus habbazi*, *Lydekkerina*, *Microposaurus*, *Paracyclotosaurus*, *Rhinesuchus* and *Xenotosuchus*) includes semi-aquatic to aquatic animals (*Schoch, 2008*). A study dedicated to *Rhinesuchus* palaeohistology concluded that it had a fully aquatic lifestyle (*McHugh, 2014*), whereas *Lydekkerina*, a basal stereospondyl, was amphibious with a tendency to be terrestrial (*Canoville and Chinsamy, 2015*).

### Pareiasaurid and bolosaurid parareptiles

The sampled parareptiles include the Chinese Bolosauridae, *Belebey chengi*, and the South African Pareiasauria, *Pareiasaurus* and basal pareiasaurs (*Embrithosaurus*, *Nochelosaurus* or

*Bradysaurus*; (*Lee, 1997*). The Chinese taxon is considered to have been terrestrial (*Berman et al., 2000*; *Müller et al., 2008*), but interpretations of the life habits and habitats of the South Africa pareiasaurs still lack a consensus. According to authors, they are variously considered fully aquatic (*Ivakhnenko, 2001*), semi-aquatic (*Kriloff et al., 2008*) or fully terrestrial (*Voigt et al., 2010*; *Canoville et al., 2014*). The low $\delta^{18}O_p$ values measured in both *Pareiasaurus* and *Rhinesuchus* are similar to each other, *Pareiasaurus* having slightly higher $\delta^{18}O_p$ values than *Rhinesuchus* (~1‰), but significantly lower than those measured in therapsids (about 4‰ to 8‰). Considering *Rhinesuchus* as an aquatic ectothermic amphibian (*McHugh, 2014*), it is suggested that the ectothermic *Pareiasaurus* was also aquatic or semi-aquatic as previously suggested (*Ivakhnenko, 2001*; *Kriloff et al., 2008*). In contrast to the semi-aquatic *Pareiasaurus* from the Lower *Daptocephalus* AZ, the *Tapinocephalus* AZ pareiasaurs may have been terrestrial as recently proposed in a study on pareiasaur ecology and based on oxygen and carbon isotope compositions of apatite carbonate (*Canoville et al., 2014*). Moreover, those authors found no significant differences between therocephalians and pareiasaurs, agreeing with our dataset, and a ~ 3‰ overlap between pareiasaur and dinocephalian $\delta^{18}O_c$ values, even though most dinocephalian values are several per mil lower than those of pareiasaurs.

## Archosauriforms

Archosauriformes are represented by the Erythrosuchidae *Erythrosuchus* and *Shansisuchus shansisuchus*, by the Proterosuchidae '*Chasmatosaurus*' *yuani*, as well as by an indeterminate basal sauropodomorph dinosaur. Proterosuchid lifestyle is still unresolved, but *Botha-Brink and Smith, 2011* favoured a terrestrial rather than an aquatic lifestyle, whereas the erythrosuchids are considered as the largest terrestrial predators of their time (*Botha-Brink and Smith, 2011*).

## Lystrosaurid therapsids

Therapsids are generally considered as terrestrial dwellers (*Kemp, 2012*) with a few uncertainties such as *Anteosaurus* considered as riparian (*Boonstra, 1955*, *Boonstra, 1962*) and *Lystrosaurus* considered by some as semi-aquatic (*Germain and Laurin, 2005*; *Ray et al., 2005*; *Botha-Brink and Angielczyk, 2010*; *Canoville and Laurin, 2010*).

Because the *Lystrosaurus-Lydekkerina* $\delta^{18}O_p$ difference falls within the same range as the *Kannemeyeria-Xenotosuchus* and *Kannemeyeria-Microposaurus* values from the *Cynognathus* subzone B assemblage, *Lystrosaurus* could be considered as having the same thermophysiology and lifestyle as *Kannemeyeria*, i.e. a terrestrial endotherm (*Figure 2A*). Because *Lystrosaurus* may have either been terrestrial (*Botha and Smith, 2006*) or semi-aquatic (*Canoville and Laurin, 2010*), it cannot be totally excluded that *Lystrosaurus* shared a similar lifestyle and thermometabolism with the amphibian *Lydekkerina*. If *Lystrosaurus* was indeed semi aquatic, then it could also be predicted to have been an ectotherm.

Given that the thermophysiology of the terrestrial archosauriform '*Chasmatosaurus*' remains unclear, the two similar $\delta^{18}O_p$ values observed in the lower Triassic Jiucaiyuan Fm. of Xinjiang province of China may indicate either that both the lystrosaurid and the proterosuchid shared similar thermometabolism and lifestyle (terrestrial endotherms or terrestrial ectotherms; *Figure 2*), or that one of them may have had a lower body temperature. Indeed, compared to low latitudes where ectotherms have slightly lower $\delta^{18}O_p$ values than co-existing endotherms due to differences in metabolic activity and body temperatures, mid latitude ectotherms have even lower body temperatures (reflecting the environmental thermal gradient) resulting in higher $\delta^{18}O_p$ values that mimic those of co-existing endotherms (*Amiot et al., 2004*). It is worth noting that based on histological features, *Proterosuchus*, had intermediate growth rates, suggesting an intermediate

thermometabolism (*Botha-Brink and Smith, 2011*) that could apply to the Chinese '*Chasmatosaurus*'. Considering a terrestrial lifestyle for the Chinese lystrosaurids (see above), and the conflicting hypotheses of *Lystrosaurus* from South Africa (terrestrial endotherm vs. semi-aquatic ectotherm), the hypothesis for terrestrial endothermy for the lystrosaurids is favoured.

