## [Decision Letter]

[Editors’ note: a previous version of this study was rejected after peer review, but the authors submitted for reconsideration. The first decision letter after peer review is shown below.]

Thank you for submitting your work entitled "Oxygen isotopes suggest elevated thermometabolism within multiple Permo-Triassic therapsid clades" for consideration by *eLife*. Your article has been reviewed by three peer reviewers, and the evaluation has been overseen by a Reviewing Editor and a Senior Editor.

Our decision has been reached after consultation between the reviewers. Based on these discussions and the individual reviews below, we regret to inform you that your work will not be considered further for publication in *eLife*.

You will see from the reviews that they were generally very positive about the question and the approach. However, there is still a considerable list of uncertainties that are listed in the reviews, especially in review #3. Resolving these would likely require more than the three-month that we allow for revisions. But we will happy to look at a new submission, in case you address all relevant points at a later time.

*Reviewer #1:*

This is a potentially important paper, presenting a new geochemical means to identify thermoreguylatory principles in early reptiles, and identifying evidence that endothermy may have arisen twice in the synapsids.

However, it is not ready for review in a journal yet as it reads like a thesis chapter. First, the different sections of the paper must be discriminated – 'Methods' are mixed through the Materials and methods and Results sections, and must be separated. By reorganizing the text and focusing on a single (rather than multiple) Materials and methods sections, the paper can become simpler, shorter, and easier to read. Make sure the Results section reports simply the results.

I'm not sure that *eLife* format has been strictly followed either – the authors should check section numbering and styles are correctly followed.

In the figures, try to minimize abbreviations (e.g. ‘Terr. Vs. semiaq. Similar thermophy.'; this is not necessary. Use pure abbreviations (e.g. AZ) or spell out in full.

Results subsection “Robustness of the stable isotope record”: This reads pretty much like 'Materials and methods', so move it back to that section.

Results section second paragraph: This paragraph is a curious mixture of Methods and Figure caption. Return 'methods' stuff to the Materials and methods section – 'were calculated' 'were compared' – all methods. Then the description of symbols and shading in Figure 2–Figure 4 should be placed in the figure captions. Then, under Results, we need a plain writing description of what the plots show.

I think 'Results' begins at subsection “Therapsid thermophysiology” – so move text, combine sensibly, and re-label sections.

No, not really: the first paragraph of subsection “Therapsid thermophysiology” is more 'Methods' – telling us which taxa were studied and from where. Please apply these general principles of scientific paper writing properly and reorganize and rationalize the paper.

*Reviewer #2:*

General comments: This paper uses comparisons of stable oxygen isotope ratios of Permian-Jurassic non-mammalian synapsids to their contemporaries to try to infer when an endothermic metabolism arose in the lineage leading to mammals. The results are thought provoking, but I think a number of changes and additions will need to be made before the paper is ready to accept. I've made a number of comments and suggestions below, but a few general issues that must be addressed:

1) I am not an isotope geochemist, and I think it will be important for one to examine the paper. I'm not so worried about the analytic methods, which seem fairly standard, but I think it would be good to get a more informed perspective on the method the authors use to predict the expected magnitude of differences between ectotherms and endotherms. This is the foundation of everything else in the paper, so it is important to make sure that this is acceptable.

2) Assuming the method is acceptable, I strongly recommend that the authors conduct a sensitivity analysis using a range of plausible average temperatures and latitudinal gradients for each of their time periods of interest. Their results will be stronger if they are consistent across the plausible range of values, and it will remove any appearance of cherry-picking temperatures/gradients to get the results that they want.

3) The authors seem to play pretty fast and loose with the terms "endothermy" and "ecothermy." A particular manifestation of this is the frequency with which they use bone histology (particularly the presence of fibrolamellar bone) as a correlate of endothermy. Although this was more common in earlier work on paleohistology, more recently the field has been much more circumspect about this relationship, with much more attention placed (rightly, in my opinion) on the implications of histology for growth rates/patterns. Elevated growth rates often are correlated with "endothermy" but this is not universally true. I think the authors need to be more thoughtful in their presentation of this relationship, and they also should provide more explicit, detailed definitions of "endothermy" and "ecothermy" at the start of the paper.

4) I recommended some changes to the discussion that would explore the link between endothermy and survivorship during the Permo-Triassic Mass Extinction (PTME) that would increase the broad interest and impact of the paper. Alternatively, the authors could provide a more detailed discussion of their results in the context of previous work and current ideas on the evolution of endothermy. Either way, much of what's in the current disucssion isn't very interesting (and I say this as someone who loves the intricacies of dicynodont taxonomy) and it undersells the paper.

5) There are a couple places in the main text and supplement that could be strengthened in terms of the references that are cited.

6) The writing style of the paper is good, but it could be improved upon. I recommend having Rubidge, Smith, and Viglietti go over the final version of the paper to clean things up and make sure the idiom is appropriate.

Abstract: mammals didn't exist in the Permian, so you should say something along the lines of "endothermy evolved in the Permian ancestors of mammals."

Abstract: Cynognathia is a subclade of Eucynodontia, so I'm not sure what you mean here.

Abstract: change to "originated in Epicynodontia during the…". Also, change to middle-late Permian.

Introduction opening sentence: this sentence is quite vague and is kind of meaningless as a result. Please be more specific about particular ways in which environmental changes affect vertebrates if you want to use this kind of example.

Introduction, first paragraph: as written, this paragraph provides something of a definition of endothermy. However, because endothermy is kind of a loaded term, and has been used in different ways by different authors, I suggest that you explicitly define what you mean by it so as to avoid any confusion.

Introduction paragraph two: again, defining endothermy will help to avoid confusion about what you mean by "true endotherms."

In the same paragraph: I think you're kind of downplaying the amount of work on this subject. For example, McNab, 1978, Hillenius, 1992; 1994, Ruben, Hillenius, Kemp and Quick, 2012, Owerkowicz, Musinsky, Middleton and Crompton, 2015, all would be appropriate to cite here.

Introduction paragraph four: bony secondary palate would be a better term to use.

In the same paragraph: Dicynodonts also have a complete secondary palate, although it is formed primarily by the maxilla.

Introduction paragraph five: Owerkowicz, Musinsky, Middleton and Crompton, 2015, would be especially good to cite in the context of whether turbinates are necessary in the context of mammalian endothermy. Also Laaß (2011) presents some evidence of turbinals in *Lystrosaurus*, although I'm kind of skeptical as to whether that's actually what's visible in the scan.

Reports of fibrolamellar bone in synapsids go back farther than these studies. For example, de Ricqles reports fibroilamellar bone in some therapsids is in papers on bone histology from the 1970's.

Fibrolamellar bone is a lot more ubiquitous among therapsids than what you suggest here; if anything it's more common than not. There's even pelycosaur-grade synapsids like *Sphenacodon* and *Ophiacodon* that produce fibrolamellar bone for at least part of their ontogenies.

In addition to the fact that some non-ectothermic teterapods can produce fibrolamellar bone, and important point to make here is that the presence of fibrolamellar bone really is more a reflection of the rate at which bone tissue is being formed than endothermy in a strict sense. While the high growth rates required to produce fibrolamellar bone often are correlated with high metabolic rates, this is not always the case.

Introduction paragraph six: although the degree of resistance to diagenesis varies among tissues, with enamel being the more resistant by far.

Materials and methods section, subsection “Sample collection”: you should provide justification for why you're using these alternate dates. Also, looking at your supplementary table, it seems like some of your dates imply more precision than is actually the case. For example, as far as I know there aren't any published radiometric dates for the Burgersdorp Fm. If you're making the correlation to marine stages (and their dates) using tetrapod biostratigraphy, I would suggest caution in this case, given the recent dates from South America that have been published by Ottone et al., 2014 and Marsicano et al., 2016. In any case more information on the source of your dates should be provided.

Subsection “Robustness of the stable isotope record”: “contain” instead of “contains”.

Figure 1: the colouration of the different parts of the plot doesn't seem to be explained. I understand that the dark orange area at the top right is the area that represents diagenesis, but do the other coloured regions have some meaning?

Subsection “Robustness of the stable isotope record”: does partial preservation of the isotopic signal require different interpretation than pristine preservation?

In the same subsection: Figure 2 is repeated here. Do you mean 2A, 3A, and 4A?

Would it be better to refer to the colours instead of light and dark gray? Since the journal is primarily an online publication the colour online/grayscale printed version seems like it isn't really a concern, and referring to the colours would help readers understand the figures.

Permian Therapsids general comment: I recommend reorganizing this section a little. I think it would be most logical to present the results subdivided by time (i.e., all middle Permian things first, then early late Permian, then latest Permian). Within each time bin, you can subdivide further by geographic area.

Early-Middle Triassic Therapsids general comment: as with the Permian taxa, I think it would be most useful to subdivide the results first by time and then by geography.

Subsection “Early to Middle Triassic therapsids”: because bone histology is really more a record of growth rates, it might be better to say something along the lines of "…which is consistent with the elevated growth rates implied by the bone histology of *Erythrosuchus*."

I think the paragraph describing the results on the Chinese *Lystrosaurid* should be combined with the previous paragraph on the South African specimens. Also, *Lystrosaurus* is the only *Lystrosaurid* known from China, so I would say its safe to assume that it is *Lystrosaurus*.

In the same subsection and in Figure 3 you make it seem like it is pretty certain that *Lystrosaurus* is an endotherm. However, in the supplement, you make it seem like there's actually a fair amount of uncertainty based on questions about the ecology of *Lystrosaurus* and the metaboloic status *'Chasmatosaurus'* (which is inferred from information on growth rates inferred from bone histological data from related taxa in South Africa). I think you need to better portray this uncertainty here, and also be sure to take it into account in the Discussion section of the paper.

Subsection “Middle to Late Triassic therapsids”: contemporary instead of co-existing

Discussion section and elsewhere: although wordier, something like "the unnamed clade comprising Lystrosauridae + Kannemeyeriiformes" would make more sense.

Discussion paragraph three: "basal dicynodontoid" would work better.

Discussion paragraph five: I don't really agree that this is a valid hypothesis. You have a fair number of Permian therapsids from different clades that are reconstructed as being ecothermic, and it seems implausible that you would have so many reversals to ecothermy. In the previous paragraph, you discuss a hypothesis based on a parsimony optimization of the character on the phylogeny. That is justified: you have a clear-cut optimality criterion and you are discussing the best hypothesis under that criterion. I don't think there's any method/optimality criterion that would support the number of reversals needed to account for all so many of your Permian therapsids being reconstructed as endotherms, so the argument is much less sound. If you're doing that, then you can pretty much make up whatever hypothesis you want because there's no constraint on what could be considered plausible.

In the same paragraph: biochemical instead of biochemicals

Discussion paragraph seven: assuming that *Lystrosaurus* was indeed an endotherm (see comments above and below), these observations are especially interesting in the context of the changes in growth and life history at the PTB noted by Botha-Brink et al., 2016, among others. This would be something interesting to explore further in the discussion, as well as a more general consideration of the role endothermy played in determining survivorship during the PTME. There's a couple obvious ways to get the space needed to do this. One would be to get rid of the previous paragraph, which I don't think is needed. Likewise, you could get rid of everything from paragraph two, and almost everything in paragraph three.

Concluding remarks: this is a really stilted way to say that most of your samples preserved an original signal that had not been diagenetically altered. I think you can get rid of most of this (see comments above).

General comment on the supplement: you should reorganize so that the first section is the one describing how you came up with your theoretical ranges of isotope values for taxa with different ecologies and thermophysiologies. That's the foundation for all of the interpretation of the fossil data you present in the paper and the supplement, so it deserves to go first. If space (or editorial largesse) allows, I would even suggest moving the information on how you calculated your expected ranges into the main text.

Appendix paragraph two: change heading to "Stereospondyl amphibians." Also make this change in the first line of this paragraph. Also, does the fact it was amphibious complicate your comparisons between it and *Lystrosaurus*?

Appendix subsection “Pareiasaurid and bolosaurid parareptiles”: would the fact that *Pareisaurus* was an herbivore whereas *Rhinesuchus* was a faunivore make any difference here? I suppose the results for the older pareiasaurus would suggest not.

Appendix subsection “Lystrosaurid therapsids”: going along with my comments above about the identity of the Chinese material, I recommend changing the title of this section to "*Lystrosaurus*."

In the same section: there is a much deeper history of debate in the literature about *Lystrosaurus* potentially being semi-aquatic that would be worth citing here: Broom 1903b, 1932; Watson 1912, 1913; Williston 1914; Brink 1951; Camp 1956; Cluver 1971; Kemp 1982; Hotton 1986; King 1991; King and Cluver 1991; Germain & Laurin 2005; Ray et al. 2005; Botha-Brink and Angielczyk 2010

Your interpretation of *Lystrosaurus* as an endotherm seems a lot more uncertain here than you portray it in the text. This uncertainty has important implications for your the evolutionary scenarios you present in the Discussion section and considerations of the potential relationships between endothermy and the PTME. Therefore, I think you need to present this uncertainty more clearly in the main text (there you make it seem like *Lystrosaurus* is definitely an endotherm, which seems kind of misleading in the context of what's in the supplement).

Appendix subsection “Theroretical curves construction”: again, there doesn't seem to be any gray areas in the figures in the main text, so I would refer to the colours you use in those figures.

It would be good to have a citation for 37C being the average body temperature of terrestrial mammals.

In the same subsection: refer to colours for the main text figures.

General comment on the three time periods: estimates of global temperature and latitudinal thermal gradients obviously are not exact (especially when you're arbitrarily picking values for the latter in at least some cases). Because of this, it would be good to have more citations for where you're getting your values from, and justification for why you chose particular values/references. Likewise, it seems like it would be good to do a sensitivity analysis using a range of different (plausible) average temperatures and thermal gradients. Your results will seem a lot stronger if you consistently reconstruct your proposed endothermic taxa as endotherms across the range of plausible values than would be the case if you you have them come out as endotherms for certain combinations of values. Doing this will also avert criticisms that you are cherry-picking values to get the results you want.

*Reviewer #3:*

I have completed my review of Rey et al.'s manuscript, "Oxygen isotopes suggest elevated thermometabolism within multiple Permo-Triasic therapsid clades'. This is a somewhat challenging paper to evaluate: The manuscript is exceptionally well written – clear, concise, scholarly, rationally organized; really a pleasure to read. And I see no reason to question the significance of the samples studied or the quality of the isotopic measurements. So, on these bases alone I would say the paper should be published with modest revisions. However, I do not believe several of the assumptions that underlie the interpretation of the isotopic data set are well justified, and therefore I don't consider the study to offer a strong new constraint on the question of therapsid physiology. On the third hand (!?!), the authors have looked at one of the few geochemical properties that potentially constrain body temperatures of these organisms, and I'm a strong believer that it is always worthwhile to publish possibly significant observations. So, I would like to see this paper published but encourage the authors to do some soul searching about some of their interpretations.

The two most important issues, in my mind, are the state of preservation of the samples, and the robustness of the models that are used as a reference frame for interpreting their findings.

First, there is evidence indicating that geochemical properties of ancient bone or dentin do not consistently constrain body temperature, except in rare cases of extraordinary preservation. Both are composed of fine-grained phosphate intermixed with organics, and demonstrably undergo diagenetic changes in oxygen isotope composition on geologically short time scales. It is asserted that phosphate resists such changes, but there is plenty of data showing this rule of thumb is often violated (see Stolper et al., 2016 for an example).

The authors have tried to avoid this by filtering the database for signs of alteration, but they do so with relatively simple, indirect metrics (percent carbonate and the phosphate-carbonate fractionation). I would have found it more compelling if they could have shown through microscopy and/or minor element mapping that they were studying well-preserved fabrics. Or if they had focused on only tooth enamel, which is generally much more resistant to alteration.

Second, the models of the oxygen isotope compositions of endotherm and ectotherm d18O values are based on two variables that strike me as poorly known or known to be false:

1) The assignments of taxa to terrestrial vs. semi-aquatic ecologies are essential to interpretations here, but my reading of the appendix suggests they are often debated and the choices made here involve an element of circular reasoning (i.e., they chose the one that leads to some simpler, preferred interpretation of the oxygen isotope data). This seems to me like a case where you really need to focus on unambiguous cases (e.g., marine reptiles, frogs, etc.).

2) It is assumed that ectotherms have growth temperatures equal to mean annual air temperatures, which are approximated with a linear dependence on latitude. These are both contraindicated by environmental data. Ectothermic tetropods generally have body temperatures well in excess of mean annual air temperature, particularly during their active seasons when they likely do most of their growth. For example, a recent survey of field-measured lizard body temperatures yielded a range of ~30-35 C across a wide range of latitudes and altitudes. And the earth's atmospheric temperature gradient is relatively gentle up to 60 ˚C, after which it really drops like a shot. The combination of these factors suggests to me that the authors have over estimated the contrast between ectotherm and endotherm d18O values, by perhaps a factor of two or more. These could have been improved on using modern field data (i.e., compare similar organisms in modern environments).

[Editors’ note: what now follows is the decision letter after the authors submitted for further consideration.]

Thank you for resubmitting your work entitled "Oxygen isotopes suggest elevated thermometabolism within multiple Permo-Triassic therapsid clades" for further consideration at *eLife*. Your revised article has been favorably evaluated by Diethard Tautz (Senior editor), a Reviewing editor, and one reviewer.

The manuscript has been improved but there are some small remaining issues that need to be addressed before acceptance, as outlined below:

*Reviewer #1:*

The paper presents an unexpected result, that endothermy arose more than once among lineages of synapsids. The paper is novel in basing this result on new isotopic data measured from fossil bones. The methods are difficult to get right, but the Lécuyer laboratory at Lyons leads the world in these methods, which adds credibility to the paper.

The Abstract is not entirely clear about the key claim, that endothermy arose twice, independently, among Synapsida. If that is the claim, say so clearly. The last sentence needs revision.

The Introduction starts with a single-sentence first paragraph about environmental changes – this could be cut. Then go straight into definition and explanation of endothermy – the first sentences about endothermy need some references to justify all the claims.

Results section: thirteen = 13. Check throughout that numbers follow the usual convention, of one to ten in full, and 11 upwards as digits, except when the number is the first word in a sentence, when it is given in full.

Discussion section paragraph two: cut – pointless sentence.

Discussion paragraph seven: inherited by = inherited from.

Discussion paragraph seven: what is the purpose of this paragraph? – it wanders everywhere. Focus simply on the key point: could endothermy have evolved once only in the ancestors of the L+K clade and Epicynodontia? Discuss and reject. Move to new paragraph.

Figure 5: make the arrows larger and brighter – maybe red. This is the key of the paper to argue the phylogenetic point. I'd also mark all sampled synapsids with a coded symbol for definitely endothermic and definitely ectothermic – this will prove that ancestors shared by the K^+^L clade and Epicynodontia were ectothermic, so confirming the likelihood that endothermy arose twice, independently.

---

## [Author Response]

[Editors’ note: the author responses to the first round of peer review follow.]

*[…] Reviewer #1:*

*This is a potentially important paper, presenting a new geochemical means to identify thermoreguylatory principles in early reptiles, and identifying evidence that endothermy may have arisen twice in the synapsids.*

*However, it is not ready for review in a journal yet as it reads like a thesis chapter. First, the different sections of the paper must be discriminated – 'Methods' are mixed through the Methods and Results sections, and must be separated. By reorganizing the text and focusing on a single (rather than multiple) Methods sections, the paper can become simpler, shorter, and easier to read. Make sure the Results section reports simply the results.*

The ‘Methods’ elements from the ‘Results’ section have been moved to the corresponding section. Several sections in Materials and methods have been selected to divide several independent parts (samples, analysis, diagenetic process, model construction).

*I'm not sure that eLife format has been strictly followed either – the authors should check section numbering and styles are correctly followed.*

There is no imposed format from *eLife*. Therefore, we chose one constant format for the manuscript and the Appendix.

*In the figures, try to minimize abbreviations (e.g. ‘Terr. Vs. semiaq. Similar thermophy.'; this is not necessary. Use pure abbreviations (e.g. AZ) or spell out in full.*

The abbreviations to describe the areas have been spelled out in full in the Figure 2, Figure 3 and 4.

*Results subsection “Robustness of the stable isotope record”: This reads pretty much like 'Methods', so move it back to that section.*

This section has been moved into ‘Materials and methods’.

*Results section second paragraph: This paragraph is a curious mixture of Methods and Figure caption. Return 'methods' stuff to the Materials and methods section – 'were calculated' 'were compared' – all methods.*

The ‘Methods’ parts have been moved to the Materials and method section.

Then the description of symbols and shading in Figure 2–Figure 4 should be placed in the figure captions.

In order to avoid very long, and identical, figure captions for Figure 2, Figure 3 and Figure 4, the description of the different areas is left in its own Methods section. This information is presented in the Figures with embedded texts for each area.

*Then, under Results, we need a plain writing description of what the plots show.*

The Results section describes the range of values obtained, country by country, for each time period, and the areas where the vales were obtained.

*I think 'Results' begins at subsection “Therapsid thermophysiology” – so move text, combine sensibly, and re-label sections.*

Section moved and re-labelled.

No, not really: the first paragraph of subsection “Therapsid thermophysiology” is more 'Methods' – telling us which taxa were studied and from where. Please apply these general principles of scientific paper writing properly and reorganize and rationalize the paper.

This paragraph has been rewritten as:

“The thirteen sampled South African Permian therapsids come from three different assemblage zones (AZ) of the Beaufort Group: the lower *Tapinocephalus* AZ, the *Tropidostoma* AZ and the lower *Daptocephalus* AZ (Viglietti et al., 2016).

Oxygen isotope compositions of three therapsid genera (*Dicynodon, Diictodon* and *Oudenodon)* from the youngest assemblage zone (lower *Daptocephalus* AZ) and seven therapsid genera (*Aelurosaurus, Diictodon, Ictidosuchoides, Oudenodon, Rhachiocephalus, Tropidostoma* and a basal cynodont) from the *Tropidostoma* AZ were respectively compared with one co-existing *Rhinesuchus* and one co-existing rhinesuchid.”

In addition: Before giving the values of the isotopic differences between the therapsids and the co-existing tetrapods, it is pertinent for us to present which of these are therapsids and their locality information. This information could be included in Materials or Methods, but we consider that it will be easier to read and comprehend if the genus name and isotopic values were placed in the same paragraph instead of separated by several pages. Moreover, some of the taxa (such as *Dicynodon, Diictodon* and *Oudenodon*) are present at several localities, and need to be cited in results. For this reason we have kept this section under results.

*Reviewer #2:*

*General comments: This paper uses comparisons of stable oxygen isotope ratios of Permian-Jurassic non-mammalian synapsids to their contemporaries to try to infer when an endothermic metabolism arose in the lineage leading to mammals. The results are thought provoking, but I think a number of changes and additions will need to be made before the paper is ready to accept. I've made a number of comments and suggestions below, but a few general issues that must be addressed:*

*1) I am not an isotope geochemist, and I think it will be important for one to examine the paper. I'm not so worried about the analytic methods, which seem fairly standard, but I think it would be good to get a more informed perspective on the method the authors use to predict the expected magnitude of differences between ectotherms and endotherms. This is the foundation of everything else in the paper, so it is important to make sure that this is acceptable.*

We have now explained in the section “2.4” how the expected magnitude of differences between ectotherms and endotherms have been defined. We added the following sentence to provide more information in the text (a detailed explanation is provided in the Appendix 1): “To construct those theoretical areas, both the phosphate-water temperature scale from Lécuyer, Amoit, Touzeau and Trott, 2013 and the differences of stable oxygen compositions between mammals of various ecologies from Cerling et al., 2008 have been used.”

*2) Assuming the method is acceptable, I strongly recommend that the authors conduct a sensitivity analysis using a range of plausible average temperatures and latitudinal gradients for each of their time periods of interest. Their results will be stronger if they are consistent across the plausible range of values, and it will remove any appearance of cherry-picking temperatures/gradients to get the results that they want.*

Following the reviewer’s request, we recalculated the areas of theoretical values for the difference between an endotherm and an ectotherm by using a range of plausible thermal latitudinal gradients. All three different gradients, 0.6°C\°L, 0.4°C\°L and 0.5°C\°L, are now with an uncertainty range of 0.1°C\°L to take into account the possible variation of this thermal latitudinal gradient within the periods selected. Therefore, we added in the Appendix 1 the following sentence: “In order to take into account the possible variations of the thermal gradient within the selected periods, an interval of ± 0.1°C/°L is added for each Main text figures.”

*3) The authors seem to play pretty fast and loose with the terms "endothermy" and "ecothermy." A particular manifestation of this is the frequency with which they use bone histology (particularly the presence of fibrolamellar bone) as a correlate of endothermy. Although this was more common in earlier work on paleohistology, more recently the field has been much more circumspect about this relationship, with much more attention placed (rightly, in my opinion) on the implications of histology for growth rates/patterns. Elevated growth rates often are correlated with "endothermy" but this is not universally true.*

We acknowledged the fact that this relationship is indirect and have thus added the following sentence in the Introduction: “However, FLB also occurs in a few ectotherms such as in some turtles and crocodilians, and is absent in small mammals and passerine birds(Bouvier, 1977), showing that FLB is mostly correlated with high growth rates, which does not always correlate to high metabolic rates.”

*I think the authors need to be more thoughtful in their presentation of this relationship, and they also should provide more explicit, detailed definitions of "endothermy" and "ecothermy" at the start of the paper.*

We added the following paragraph in the Introduction:

“Endothermy is commonly associated with homeothermy, being the capacity to regulate the body heat through metabolic activity as well. This combination corresponds to one end of a gradient of thermoregulatory strategies observed in living animals. The other end of the spectrum is ectothermy combined with poïkilothermy which animals use as a thermoregulatory strategy to increase their body temperature toward optimal levels by using external heat sources. Their body temperature therefore traces that of their surroundings and is the most occurring saving strategy.”

*4) I recommended some changes to the discussion that would explore the link between endothermy and survivorship during the Permo-Triassic Mass Extinction (PTME) that would increase the broad interest and impact of the paper. Alternatively, the authors could provide a more detailed discussion of their results in the context of previous work and current ideas on the evolution of endothermy. Either way, much of what's in the current discussion isn't very interesting (and I say this as someone who loves the intricacies of dicynodont taxonomy) and it undersells the paper.*

In order to make this important part of the paper more exciting we have extensively modified it and also added the following paragraph: “A possible explanation could be the acquisition of a fast growth rate due to the high metabolic rate of the endothermy. According to a recent palaeohistology study (Botha-Brink et al., 2016) Early Triassic therapsids, such as *Lystrosaurus,* therocephalians and cynodonts, had a high growth rate allowing them to reach reproductive maturity within a few seasons and compensate their shortened life expectancies. This adaptation might have enabled certain therapsids to survive the intense climatic change of that time and conquer the newly vacant niches.”

*5) There are a couple places in the main text and supplement that could be strengthened in terms of the references that are cited.*

Numerous references have been added to address this comment.

*6) The writing style of the paper is good, but it could be improved upon. I recommend having Rubidge, Smith, and Viglietti go over the final version of the paper to clean things up and make sure the idiom is appropriate.*

All three of them went through the manuscript after all the comments from the authors were applied.

*Abstract: mammals didn’t exist in the Permian, so you should say something along the lines of “endothermy evolved in the Permian ancestors of mammals.”*

“[M]ammal” has been changed to “mammal ancestors”.

*Abstract: Cynognathia is a subclade of Eucynodontia, so I’m not sure what you mean here.*

“Cynognathia and Eucynodontia” has been changed to “Eucynodontia”.

*Abstract: change to “originated in Epicynodontia during the…”. Also, change to middle-late Permian.*

Changes have been made.

*Introduction opening sentence: this sentence is quite vague and is kind of meaningless as a result. Please be more specific about particular ways in which environmental changes affect vertebrates if you want to use this kind of example.*

We have modified the sentence as follows: “Rapid environmental changes have played a major role in shaping the evolutionary history of terrestrial vertebrates not only as the major cause of mass extinctions but also favouring species with biological adaptations that allowed them to survive and oikilothe the altered ecosystems..”

*Introduction, first paragraph: as written, this paragraph provides something of a definition of endothermy. However, because endothermy is kind of a loaded term, and has been used in different ways by different authors, I suggest that you explicitly define what you mean by it so as to avoid any confusion.*

We added the following paragraph in the Introduction: “Endothermy is commonly associated with homeothermy, being the capacity to regulate the body heat through metabolic activity as well. This combination corresponds to one end of a gradient of thermoregulatory strategies observed in living animals. The other end of the spectrum is ectothermy combined with oikilothermy which animals use as a thermoregulatory strategy to increase their body temperature toward optimal levels by using external heat sources. Their body temperature therefore traces that of their surroundings and is the most occurring saving strategy.”

*Introduction paragraph two: again, defining endothermy will help to avoid confusion about what you mean by “true endotherms.”*

The sentence has been modified to: “Amongst modern vertebrates, various thermoregulatory strategies between these two end-members have been adopted, such as regional endothermy (Bernal, Dickson, Shadwick and Graham, 2001; Katz, 2002) or inertial homeothermy (McNab and Auffenberg, 1976), and only birds and mammals fall within the endothermy-end of the spectrum.”

*In the same paragraph: I think you're kind of downplaying the amount of work on this subject. For example, McNab, 1978, Hillenius, 1992; 1994, Ruben, Hillenius, Kemp and Quick, 2012, Owerkowicz, Musinsky, Middleton and Crompton, 2015, all would be appropriate to cite here.*

This section has been rewritten and all the suggested references have been added.

*Introduction paragraph four: bony secondary palate would be a better term to use.*

The term has been changed.

*In the same paragraph: Dicynodonts also have a complete secondary palate, although it is formed primarily by the maxilla.*

We added the following sentence: “It is noteworthy that complete bony secondary palate is present in dicynodonts, the difference being that it is primarily formed by the premaxilla (King, 1988) whereas it is mostly the maxilla in therocephalians, cynodonts and extant mammals.”

*Introduction paragraph five: Owerkowicz, Musinsky, Middleton and Crompton, 2015 would be especially good to cite in the context of whether turbinates are necessary in the context of mammalian endothermy. Also Laaß (2011) presents some evidence of turbinals in Lystrosaurus, although I'm kind of skeptical as to whether that's actually what's visible in the scan.*

The first reference has been added.

*Reports of fibrolamellar bone in synapsids go back farther than these studies. For example, de Ricqles reports fibroilamellar bone in some therapsids is in papers on bone histology from the 1970's.*

We modified accordingly and added the following references: “(de Ricqlès, 1972, 1979; Botha, 2003; Botha and Chinsamy, 2001, 2004; Ray, Botha and Chinsamy, 2004; Olivier, Houssaye, Jalil and Cubo, 2017)”.

*Fibrolamellar bone is a lot more ubiquitous among therapsids than what you suggest here; if anything it's more common than not. There's even pelycosaur-grade synapsids like Sphenacodon and Ophiacodon that produce fibrolamellar bone for at least part of their ontogenies.*

We modified the paragraph to: “Accordingly, several bone palaeohistological studies have have addressed the quest for the presence of FLB in therapsids (de Ricqlès, 1972, 1979; Botha, 2003; Botha and Chinsamy, 2001, 2004; Ray, Notha and Chinsamy, 2004; Olivier, Houssaye, Jalil and Cubo, 2017). Ray, Notha and Chinsamy, 2004 and Olivier, Houssaye, Jalil and Cubo, 2017 analysed several therapsid groups (anomodont, gorgonopsian, therocephalian, cynodont) and found FLB in some genera (*Aelurognathus, Pristerognathus, Tritylodon, Oudenodon, Lystrosaurus, Moghreberia*), suggesting sustained fast growth and thus elevated metabolic activity. The presence of FLB has also been demonstrated in some earlier non-therapsid synapsids such as *Sphenacodon, Dimetrodon* or even *Ophiacodon* (Huttenlocker, Rega and Sumida, 2010; Shelton, Sander, Stein and Winkelhorst, 2012; Shelton and Sander, 2015).”

*In addition to the fact that some non-ectothermic teterapods can produce fibrolamellar bone, and important point to make here is that the presence of fibrolamellar bone really is more a reflection of the rate at which bone tissue is being formed than endothermy in a strict sense. While the high growth rates required to produce fibrolamellar bone often are correlated with high metabolic rates, this is not always the case.*

We have added the following sentence: “However, FLB also occurs in a few ectotherms such as in some turtles and crocodilians, and is absent in small mammals and passerine birds(Bouvier, 1977), showing that FLB is mostly correlated with high growth rates, which does not always correlate to high metabolic rates.”

*Introduction paragraph six: although the degree of resistance to diagenesis varies among tissues, with enamel being the more resistant by far.*

The paragraph here explains the reason why the oxygen isotopic composition of the phosphate group is used. The difference between tissues is described later in “Materials and methods” as follows: “Analysed materials consist of bone or tooth dentine, which is more porous than enamel with small and less densely inter-grown apatite crystals (Mills, 1967)”.

*Materials and methods section, subsection “Sample collection”: you should provide justification for why you're using these alternate dates.*

Since the submission of this manuscript, the alternative dates have become the official ones. We have modified the sentence to read: “All the sample localities have been correlated to the marine biostratigraphic stages using the absolute ages accepted by the International Commission on Stratigraphy (Cohen et al., 2013; updated 12/2016), with the Permo-Triassic and Guadalupian-Lopingian boundaries now considered to be at 251.90 ± 0.02 Ma (Burgess, Bowring and Shen, 2014) and 259.1 ± 0.5 (Zhong, He, Mundil and Xu, 2014) Ma, respectively.”

*Also, looking at your supplementary table, it seems like some of your dates imply more precision than is actually the case. For example, as far as I know there aren't any published radiometric dates for the Burgersdorp Fm. If you're making the correlation to marine stages (and their dates) using tetrapod biostratigraphy, I would suggest caution in this case, given the recent dates from South America that have been published by Ottone et al. (2014) and Marsicano et al. (2016). In any case more information on the source of your dates should be provided.*

We intended to give an age value to every studied stratigraphic interval. However, as pointed out by the reviewer, not all formations have radiometric ages and in these instances we used biostratrigraphic correlation (and references) as explained in the text:

“Triassic age determination has been achieved by biostratigraphic correlation with Laurasian sequences (Hancox, Shishkin, Rubidge and Kitching, 1995; Rubidge, 2005; Abdala and Ribeiro, 2010).”

“The locality belongs to the “Red Beds inférieurs a or b” of the lower Elliot Formation which is currently regarded as latest Triassic (late Rhaetian) (Knoll, 2004).” “The locality is biostratigraphically correlated to the last segment of the Timezgadiouine Formation, considered to be Middle to early Late Carnian(Jalil, 1999).”

“The Dashankou locality, from Gansu Province, is biostratigraphically dated as Early Roadian (Liu and Rubidge, 2009; Liu, 2010). From Shanxi Province, sampled fossils originate from three localities in the Ermaying Formation which is considered to be Anisian (Liu, Li and Li, 2013). From Xinjiang Province, two localities in the Jiucaiyuan Formation have been sampled and are considered Early Triassic (Metcalfe et al., 2009).”

*Subsection “Robustness of the stable isotope record”: “contain” instead of “contains”.*

The modification has been made.

*Figure 1: The colouration of the different parts of the plot doesn't seem to be explained. I understand that the dark orange area at the top right is the area that represents diagenesis, but do the other coloured regions have some meaning?*

The arrows on the side describe the figure with all the upper section corresponding to phosphate diagenesis, and the part on the right corresponding to carbonate diagenesis. Therefore, the upper right part corresponds to both phosphate and carbonate diagenesis.

*Subsection “Robustness of the stable isotope record”: does partial preservation of the isotopic signal require different interpretation than pristine preservation?*

Partial preservation means that some of the samples might have their isotopic compositions slightly modified by diagenesis, meaning that it is impossible to be 100% certain if the signal is completely original. With no way to quantify it, we interpret our data as preserved, acknowledging the possible slight alteration. Therefore, we modified the sentence as follows: “Using these two assessments, newly measured δ^18^O_p_ values are considered to have preserved their original isotopic signatures and can be interpreted in terms of ecologies and physiologies”.

*In the same subsection: Figure 2 is repeated here. Do you mean 2A, 3A, and 4A?*

The modification has been made to 2A, 3A and 4A.

*Would it be better to refer to the colours instead of light and dark gray? Since the journal is primarily an online publication the colour online/grayscale printed version seems like it isn't really a concern, and referring to the colours would help readers understand the figures.*

We have now included colour as suggested and all the shades of grey have been changed to the corresponding colours.

*Permian Therapsids general comment: I recommend reorganizing this section a little. I think it would be most logical to present the results subdivided by time (i.e., all middle Permian things first, then early late Permian, then latest Permian). Within each time bin, you can subdivide further by geographic area.*

The chosen subdivision has been made in order to avoid repetition and make reading easier. Accordingly we first listed the South African fossils because of their greater numbers and their possible occurrences in several AZ (eg. *Diictodon* and *Oudenodon* in both lower *Daptocephalus* AZ and *Tropidostoma* AZ). Some AZ are also linked together with the groups used for comparison (eg. stereospondyls).

*Early-Middle Triassic Therapsids general comment: as with the Permian taxa, I think it would be most useful to subdivide the results first by time and then by geography.*

In this case we preferred to subdivide the results first by country with a link between South Africa and China, which included *Lystrosaurus* which is common to both. Division by time would cause the text to alternate between the South African and Chines fossils.

*Subsection “Early to Middle Triassic therapsids”: because bone histology is really more a record of growth rates, it might be better to say something along the lines of "…which is consistent with the elevated growth rates implied by the bone histology of Erythrosuchus."*

The sentence has been changed to: “*Erythrosuchus* was also endothermic which is consistent with the elevated growth rates implied by its palaeohistology”.

*I think the paragraph describing the results on the Chinese lystrosaurid should be combined with the previous paragraph on the South African specimens. Also, Lystrosaurus is the only Lystrosaurid known from China, so I would say its safe to assume that it is Lystrosaurus.*

As suggested the paragraphs have been combined. As there is no more precise official identification for the Chinese fossil, we preferred to leave it as it is.

*In the same subsection and in Figure 3 you make it seem like it is pretty certain that Lystrosaurus is an endotherm. However, in the supplement, you make it seem like there's actually a fair amount of uncertainty based on questions about the ecology of Lystrosaurus and the metaboloic status 'Chasmatosaurus' (which is inferred from information on growth rates inferred from bone histological data from related taxa in South Africa). I think you need to better portray this uncertainty here, and also be sure to take it into account in the Discussion section of the paper.*

The Appendix paragraph mentions that the South African *Lystrosaurus* is either terrestrial-endotherm or semiaquatic-ectotherm, whereas the Chinese one is either terrestrial endotherm or terrestrial ectotherm. Considering that the South African and Chinese *Lystrosaurus* probably occupied the same ecological niche, the only common possibility is terrestrial endotherm.

*Subsection “Middle to Late Triassic therapsids”: contemporary instead of co-existing*

Modification has been made.

*Discussion section and elsewhere: although wordier, something like "the unnamed clade comprising Lystrosauridae + Kannemeyeriiformes" would make more sense.*

The sentence has been modified to: “the unnamed dicynodont clade comprising Lystrosauridae + Kannemeyeriiformes, abbreviated the “L+K” clade later,”

*Discussion paragraph three: "basal dicynodontoid" would work better.*

Modification has been made.

*Discussion paragraph five: I don't really agree that this is a valid hypothesis. You have a fair number of Permian therapsids from different clades that are reconstructed as being ecothermic, and it seems implausible that you would have so many reversals to ecothermy. In the previous paragraph, you discuss a hypothesis based on a parsimony optimization of the character on the phylogeny. That is justified: you have a clear-cut optimality criterion and you are discussing the best hypothesis under that criterion. I don't think there's any method/optimality criterion that would support the number of reversals needed to account for all so many of your Permian therapsids being reconstructed as endotherms, so the argument is much less sound. If you're doing that, then you can pretty much make up whatever hypothesis you want because there's no constraint on what could be considered plausible.*

We consider this second hypothesis still valid, as it does not involve reversal from endothermy to ectothermy, but that the characteristics for endothermy were present but not a full endothermy. Therefore, we added the following sentences: “This suggests that biochemical and physiological mechanisms allowing mammal endothermy, appeared at the base of the neotherapsids at any given time during the middle Permian which is a conclusion recently published based on the paleohistology of some dicynodonts (Olivier, Houssaye, Jalil and Cubo, 2017). In our case, the absence of an endothermic signal in the sampled Permian therapsids could be due to an endothermy being only seasonal, linked to a cold season or to the reproduction period (as observed today in some reptile species; Tattersall et al., 2016), which would not be visible in a bulk signal. Therefore, effective acquisition of mammal “true endothermy” was expressed within these two lineages, possibly as a result of extrinsic factors.”

*In the same paragraph: biochemical instead of biochemicals*

Modification has been made.

*Discussion paragraph seven: assuming that Lystrosaurus was indeed an endotherm (see comments above and below), these observations are especially interesting in the context of the changes in growth and life history at the PTB noted by Botha-Brink et al., 2016, among others. This would be something interesting to explore further in the discussion, as well as a more general consideration of the role endothermy played in determining survivorship during the PTME. There's a couple obvious ways to get the space needed to do this. One would be to get rid of the previous paragraph, which I don't think is needed. Likewise, you could get rid of everything from paragraph two, and almost everything in paragraph three.*

We included this hypothesis at the end of the Discussion: “A possible explanation could be the acquisition of a fast growth rate due to the high metabolic rate of the endothermy. According to a recent palaeohistology study (Botha-Brink et al., 2016) Early Triassic therapsids, such as *Lystrosaurus,* therocephalians and cynodonts, had a high growth rate allowing them to reach reproductive maturity within a few seasons and compensate their shortened life expectancies. This adaptation might have enabled certain therapsids to survive the intense climatic change of that time and conquer the newly vacant niches.”

*Concluding remarks: this is a really stilted way to say that most of your samples preserved an original signal that had not been diagenetically altered.*

We deleted the offensive sentence and modified the second one as it follows: “Assuming that analysed samples have preserved their original isotope composition of phosphate, all the Permian therapsids analysed appear to have ectotherm-like thermoregulation and representatives of two Triassic therapsid clades are considered to have had endotherm-like thermoregulation: the Lystrosauridae + Kannemeyeriiformes and the Eucynodontia.”

*I think you can get rid of most of this (see comments above).*

We think that this hypothesis is still valid (see above).

*General comment on the supplement: you should reorganize so that the first section is the one describing how you came up with your theoretical ranges of isotope values for taxa with different ecologies and thermophysiologies. That's the foundation for all of the interpretation of the fossil data you present in the paper and the supplement, so it deserves to go first. If space (or editorial largesse) allows, I would even suggest moving the information on how you calculated your expected ranges into the main text.*

The section “Theoretical curve construction” has been moved to the first section. However, to keep the “Materials and methods” section as simple as possible, we deliberately chose to keep the calculation of the theoretical range in the Appendix. We have now added the following sentence in the main text: “To construct those theoretical areas, both the phosphate-water temperature scale from Lécuyer, Amoit, Touzeau and Trotter, 2013 and the differences of stable oxygen compositions between mammals of various ecologies from Cerling et al., 2008, have been used.”

*Appendix paragraph two: change heading to "Stereospondyl amphibians." Also make this change in the first line of this paragraph.*

Changes have been made.

*Also, does the fact it was amphibious complicate your comparisons between it and Lystrosaurus?*

Not really, the Figure 2, Figure 3 and Figure 4 are the comparison between a therapsid and an amphibious animal.

*Appendix subsection “Pareiasaurid and bolosaurid parareptiles”: would the fact that Pareisaurus was an herbivore whereas Rhinesuchus was a faunivore make any difference here? I suppose the results for the older pareiasaurus would suggest not.*

The effect of diet on the oxygen isotope ratio is not overwhelming with regard to the importance of the water budget. Differences in diet may produce a much more significant difference for carbon and nitrogen isotope ratios but nothing significant for the oxygen.

*Appendix subsection “Lystrosaurid therapsids”: going along with my comments above about the identity of the Chinese material, I recommend changing the title of this section to "Lystrosaurus."*

As mentioned above: Even though we agree with the comment of the reviewer, we prefer to retain the term “Chinese lystrosaurid” until an official new identification has been made.

*In the same section: there is a much deeper history of debate in the literature about Lystrosaurus potentially being semi-aquatic that would be worth citing here: Broom 1903b, 1932; Watson 1912, 1913; Williston 1914; Brink 1951; Camp 1956; Cluver 1971; Kemp 1982; Hotton 1986; King 1991; King and Cluver 1991; Germain & Laurin 2005; Ray et al. 2005; Botha-Brink and Angielczyk 2010*

The references from 2005 to present have been added in order to present that this point of view. Several authors are still debating the subject.

*Your interpretation of Lystrosaurus as an endotherm seems a lot more uncertain here than you portray it in the text. This uncertainty has important implications for your the evolutionary scenarios you present in the Discussion section and considerations of the potential relationships between endothermy and the PTME. Therefore, I think you need to present this uncertainty more clearly in the main text (there you make it seem like Lystrosaurus is definitely an endotherm, which seems kind of misleading in the context of what's in the supplement).*

As mentioned above: The Appendix paragraph mentions that the South African *Lystrosaurus* is either terrestrial-endotherm or semiaquatic-ectotherm, whereas the Chinese one is either terrestrial endotherm or terrestrial ectotherm. Considering that the South African and Chinese *Lystrosaurus* probably occupied the same ecological niche, the only common possibility is terrestrial endotherm. We do not think that the text is misleading our interpretation of the ecology of *Lystrosaurus.*

*Appendix subsection “Theroretical curves construction”: again, there doesn't seem to be any gray areas in the figures in the main text, so I would refer to the colours you use in those figures.*

We replaced all the shades with their respective colours.

*It would be good to have a citation for 37C being the average body temperature of terrestrial mammals.*

A reference has been added as follows: “For endothermic vertebrates, T_b_ is assumed to be 37°C, the average T_b_ of most extant placentals and possibly of the common ancestor of all extant mammals (Watson and Graves, 1988).

*In the same subsection: refer to colours for the main text figures.*

We changed all the shades to their respective colours.

*General comment on the three time periods: estimates of global temperature and latitudinal thermal gradients obviously are not exact (especially when you're arbitrarily picking values for the latter in at least some cases). Because of this, it would be good to have more citations for where you're getting your values from, and justification for why you chose particular values/references. Likewise, it seems like it would be good to do a sensitivity analysis using a range of different (plausible) average temperatures and thermal gradients. Your results will seem a lot stronger if you consistently reconstruct your proposed endothermic taxa as endotherms across the range of plausible values than would be the case if you you have them come out as endotherms for certain combinations of values. Doing this will also avert criticisms that you are cherry-picking values to get the results you want.*

In order to reinforce our interpretations, we extended our theoretical ranges by adding an uncertainty of ± 0.1°C/°L for each thermal gradient we used. Uncertainties may reflect the possible variations of temperature during each chosen timespan. No interpretations were affected by this increase in those theoretical ranges.

*Reviewer #3:*

*I have completed my review of Rey et al.'s manuscript, "Oxygen isotopes suggest elevated thermometabolism within multiple Permo-Triasic therapsid clades'. This is a somewhat challenging paper to evaluate: The manuscript is exceptionally well written – clear, concise, scholarly, rationally organized; really a pleasure to read. And I see no reason to question the significance of the samples studied or the quality of the isotopic measurements. So, on these bases alone I would say the paper should be published with modest revisions. However, I do not believe several of the assumptions that underlie the interpretation of the isotopic data set are well justified, and therefore I don't consider the study to offer a strong new constraint on the question of therapsid physiology. On the third hand (!?!), the authors have looked at one of the few geochemical properties that potentially constrain body temperatures of these organisms, and I'm a strong believer that it is always worth while to publish possibly significant observations. So, I would like to see this paper published but encourage the authors to do some soul searching about some of their interpretations.*

*The two most important issues, in my mind, are the state of preservation of the samples, and the robustness of the models that are used as a reference frame for interpreting their findings.*

*First, there is evidence indicating that geochemical properties of ancient bone or dentin do not consistently constrain body temperature, except in rare cases of extraordinary preservation. Both are composed of fine-grained phosphate intermixed with organics, and demonstrably undergo diagenetic changes in oxygen isotope composition on geologically short time scales. It is asserted that phosphate resists such changes, but there is plenty of data showing this rule of thumb is often violated (see Stolper et al., 2016 for an example).*

*The authors have tried to avoid this by filtering the database for signs of alteration, but they do so with relatively simple, indirect metrics (percent carbonate and the phosphate-carbonate fractionation). I would have found it more compelling if they could have shown through microscopy and/or minor element mapping that they were studying well-preserved fabrics. Or if they had focused on only tooth enamel, which is generally much more resistant to alteration.*

Admittedly enamel would be the best apatite to sample, but before its development in the Mammalia, enamel was just a thin layer and was possibly not sufficient to sample for the analysis. Moreover, some species did not have enamel at all (eg. *Diictodon*), or even teeth (eg. *Oudenodon*), therefore dentine and bone offer the best alternative. Much of the data presented in this manuscript has already been published (Rey et al., 2016) and showed a correlation with marine data from South China through the studied interval. Currently there is no method to demonstrate if diagenetic processes might have significantly altered the oxygen isotope compositions of the phosphate group. In their study, Pucéat et al., 2004 showed that recrystallization and Rare Earth Element fractionations are possible without altering oxygen isotope compositions.

*Second, the models of the oxygen isotope compositions of endotherm and ectotherm d18O values are based on two variables that strike me as poorly known or known to be false:*

*1) The assignments of taxa to terrestrial vs. semi-aquatic ecologies are essential to interpretations here, but my reading of the appendix suggests they are often debated and the choices made here involve an element of circular reasoning (i.e., they chose the one that leads to some simpler, preferred interpretation of the oxygen isotope data). This seems to me like a case where you really need to focus on unambiguous cases (e.g., marine reptiles, frogs, etc.).*

*2) It is assumed that ectotherms have growth temperatures equal to mean annual air temperatures, which are approximated with a linear dependence on latitude. These are both contraindicated by environmental data. Ectothermic tetropods generally have body temperatures well in excess of mean annual air temperature, particularly during their active seasons when they likely do most of their growth. For example, a recent survey of field-measured lizard body temperatures yielded a range of ~30-35 C across a wide range of latitudes and altitudes. And the earth's atmospheric temperature gradient is relatively gentle up to 60 ˚C, after which it really drops like a shot. The combination of these factors suggests to me that the authors have over estimated the contrast between ectotherm and endotherm d18O values, by perhaps a factor of two or more. These could have been improved on using modern field data (i.e., compare similar organisms in modern environments).*

Comparing isotope compositions to understand thermophysiology of fossil species have been already published a few times (Amiot et al., 2006; Barrick et al., 1994; 1996). Here, we used a theoretical framework based on extant species to increase the validity of our result. Nevertheless, we added a paragraph in the Discussion where we acknowledge the possible limits of this theoretical framework: “Such interpretations are based on a theoretical framework in which estimations had been made as accurately as possible, but may not entirely represent the complex ecologies or thermophysiologies of each analysed tetrapod or the environmental conditions where they lived. Therefore, even though we have confidence in these results, it is important to compare them with other palaeobiological proxies to increase their validity.”

[Editors' note: the author responses to the re-review follow.]

*Reviewer #1:*

*The paper presents an unexpected result, that endothermy arose more than once among lineages of synapsids. The paper is novel in basing this result on new isotopic data measured from fossil bones. The methods are difficult to get right, but the Lécuyer laboratory at Lyons leads the world in these methods, which adds credibility to the paper.*

*The Abstract is not entirely clear about the key claim, that endothermy arose twice, independently, among Synapsida. If that is the claim, say so clearly. The last sentence needs revision.*

We clarified this as follows:

“It is proposed that cynodonts and dicynodonts independently acquired constant elevated thermometabolism, respectively within the Eucynodontia and Lystrosauridae + Kannemeyeriiformes clades”.

The sentence has been changed to:

“We conclude that mammalian endothermy originated in the Epicynodontia during the middle-late Permian. Major global climatic and environmental fluctuations were the most likely selective pressures on the success of such elevated thermometabolism.”

*The Introduction starts with a single-sentence first paragraph about environmental changes – this could be cut.*

The sentence has been removed.

*Then go straight into definition and explanation of endothermy – the first sentences about endothermy need some references to justify all the claims.*

Several sentences had references added:

“One key adaptation enabling tetrapods to cope with fluctuating climatic conditions was the acquisition of endothermy (Paaijmans et al., 2013). This character is defined here as the ability to actively produce body heat through metabolic activity (Cannon and Nedergaard, 2004).”

“…endothermic vertebrates are able to colonise environments with extreme thermal conditions, for example freezing at high latitudes and altitudes (Khaliq et al., 2014).”

*Results section: thirteen = 13. Check throughout that numbers follow the usual convention, of one to ten in full, and 11 upwards as digits, except when the number is the first word in a sentence, when it is given in full.*

The modifications have been applied.

*Discussion section paragraph two: cut – pointless sentence.*

The sentence has been removed.

*Discussion paragraph seven: inherited by = inherited from.*

The modification has been made.

*Discussion paragraph seven: what is the purpose of this paragraph? – it wanders everywhere. Focus simply on the key point: could endothermy have evolved once only in the ancestors of the L+K clade and Epicynodontia? Discuss and reject. Move to new paragraph.*

This paragraph shows that even with an independent acquisition of endothermy in the two clades, we cannot discard a possible homologous origin of the metabolic process leading to constant endothermy in some clades. Accordingly we modified the last sentence of that paragraph, as follows:

“Therefore, effective acquisition of mammal “true endothermy” was expressed independently within these two lineages, possibly as a result of extrinsic factors.”

*Figure 5: make the arrows larger and brighter – maybe red. This is the key of the paper to argue the phylogenetic point. I'd also mark all sampled synapsids with a coded symbol for definitely endothermic and definitely ectothermic – this will prove that ancestors shared by the K^+^L clade and Epicynodontia were ectothermic, so confirming the likelihood that endothermy arose twice, independently.*

The arrows have been modified to be larger and red. The endothermic species were already in bold, and are now also in red.